# Universal non-Hermitian skin effect in two and higher dimensions

Kai Zhang [1,2], Zhesen Yang [3✉] & Chen Fang [1,3,4✉]

Skin effect, experimentally discovered in one dimension, describes the physical phenomenon that on an open chain, an extensive number of eigenstates of a non-Hermitian Hamiltonian are localized at the end(s) of the chain. Here in two and higher dimensions, we establish a theorem that the skin effect exists, if and only if periodic-boundary spectrum of the Hamiltonian covers a finite area on the complex plane. This theorem establishes the universality of the effect, because the above condition is satisfied in almost every generic non-Hermitian Hamiltonian, and, unlike in one dimension, is compatible with all point-group symmetries. We propose two new types of skin effect in two and higher dimensions: the corner-skin effect where all eigenstates are localized at corners of the system, and the geometry-dependent-skin effect where skin modes disappear for systems of a particular shape, but appear on generic polygons. An immediate corollary of our theorem is that any non-Hermitian system having exceptional points (lines) in two (three) dimensions exhibits skin effect, making this phenomenon accessible to experiments in photonic crystals, Weyl semimetals, and Kondo insulators.

[1] Beijing National Laboratory for Condensed Matter Physics, and Institute of Physics, Chinese Academy of Sciences, 100190 Beijing, China. [2] University of Chinese Academy of Sciences, 100049 Beijing, China. [3] Kavli Institute for Theoretical Sciences, Chinese Academy of Sciences, 100190 Beijing, China. [4] Songshan Lake Materials Laboratory, Dongguan, Guangdong 523808, China. ✉email: yangzs@ucas.ac.cn; cfang@iphy.ac.cn

The study of non-Hermitian Hamiltonians, which can be regarded as the effective description of dissipative processes, can be traced back to the investigation of alpha decay, where real and imaginary parts of the complex energy are related to the experimentally observed energy level and decay rate[1]. When a lattice system is coupled with environments and has dissipations, e.g. photonic crystals having radiational loss[2–4] and electronic systems having finite quasiparticle lifetime[5,6], the non-Hermitian band theory becomes a conceptually simple and efficient approach[7–12].

Skin effect[13–23], a phenomenon unique to the non-Hermitian band theory, refers to the localization of eigenstates at the boundary, the number of which scales with the volume of the system. For example, in one dimension, all eigenstates of a non-Hermitian Hamiltonian can be localized at the ends of a chain[13]. This suggests the failure of Bloch's theorem[24,25], which states that eigenstates in the bulk are modulated plane waves. As Bloch's theorem plays a fundamental role in the development of condensed-matter physics[26], the emergence of skin effect indicates a new and possibly revolutionary direction. Especially, the skin effect has been observed experimentally in one-dimensional classical systems[27–29], inspiring further studies on their higher dimensional generalizations[14,30–37]. However, a general theory for the higher-dimensional skin effect has not been established.

Apart from the skin effect, another focus topic in non-Hermitian band systems is the exceptional point (or line)[38–47] that refers to stable point-type (or line-type) non-Hermitian band degeneracy in the Brillouin zone. At the exceptional point, not only eigenvalues but also eigenstates of the Bloch Hamiltonian coalesce[39]. Many a novel phenomenon related to exceptional points has been predicted and observed[47–52], such as the emergence of bulk-Fermi arc terminated at the exceptional points[5,45]. Since the bulk-boundary correspondence plays a central role in the development of topological phases[53], it is natural to ask if there exists a non-Hermitian bulk-boundary correspondence in bands having exceptional points, analogous to the surface Fermi arc in the Weyl semimetals in the Hermitian counterpart[54].

In this paper, we establish a theorem that reveals a universal bulk-boundary correspondence in two and higher dimensional non-Hermitian bands, as shown in Fig. 1. The "bulk" refers to the area of the spectrum of the Hamiltonian on the complex plane with periodic boundary condition, and "boundary" means the presence (absence) of the skin effect for open-boundary system of a generic shape. The theorem states that the skin effect appears if and only if the spectral area is nonzero. This skin effect is "universal" for three reasons: (i) a randomly generated local non-Hermitian Hamiltonian has the skin effect with probability one; (ii) the skin effect is, unlike in one dimension, compatible with all point-group symmetries and time-reversal symmetry, including complex-conjugate-type and transpose-type time-reversal symmetry in ref. [11]; and (iii) it does not require any special geometry of the open-boundary system. We classify the universal skin effect into non-reciprocal skin effect and generalized reciprocal skin effect according to nonzero and zero current functional, respectively, and also propose the corner-skin effect and geometry-dependent-skin effect as representative phenomena of these two categories.

A surprising corollary of our theorem is that the stable exceptional points[8,41,43] imply the presence of skin effect. Because exceptional points have been either observed or proposed in meta-materials as well as in condensed matter, this corollary makes skin effect observable in known systems. We predict the geometry-dependent skin effect in the two-dimensional photonic crystal studied in ref. [45], and propose to observe this effect in the anomalous dynamics of wave packets.

## Results

### Theorem: an equivalence between spectral area and skin effect.

For generic one-dimensional non-Hermitian systems, the correspondence between the spectral shape and the skin effect has been derived[17,18], i.e., when the Bloch spectrum is a loop-type (an arc-type), the skin effect appears (disappears).

Generalizing the correspondence to two dimensions, we note two main differences. One difference is in the periodic-boundary spectrum, $E_i(\mathbf{k})$, where $i$ is the band index and $\mathbf{k}$ the crystal momentum in the first Brillouin zone (BZ). Generally speaking, $E_i(\mathbf{k})$ is a mapping from the $d$-dimensional torus to the complex plane, $\mathbb{C}$. When $d = 1$, the image of $E_i(k)$ forms a loop; but when $d > 1$, the image is generically a continuum on $\mathbb{C}$, denoted by $E_i(\mathrm{BZ})$. The area covered by $E_i(\mathrm{BZ})$ on the complex plane is called the *spectral area*, denoted by $A_i$. Another difference is in the variety of open-boundary condition. There is only one geometry for an open system in one dimension, i.e., an open chain; but there are an infinite number of geometries in two dimensions such as triangle, rectangle and pentagon.

Now we are ready to state the theorem of universal skin effect: in the thermodynamic limit, the skin effect is present in a Hamiltonian having open boundary of generic geometry, if the spectral area is nonzero ($A_i \neq 0$); vice versa, the skin effect is absent for all possible geometries, if the spectral area is zero ($A_i = 0$). The open boundary in the theorem refers to the fully open boundary condition in all spatial directions. As the periodic-boundary Hamiltonian describes the dynamics in the bulk, the theorem relates a bulk property (spectral area) to a boundary one (existence of skin modes). Fig. 1 shows some schematic examples. The complete proof of the theorem is provided in the Supplementary Note 1.

A brief outline of the proof is illustrated in Fig. 2. The theorem is obtained in three steps: step I establishes the equivalence relation between spectral area and spectral winding number of straight lines in the BZ; step II connects these nonzero spectral winding numbers with skin effect on the stripe geometry — the geometry with open boundary in only one direction and periodic boundary in other directions; step III illustrates that skin effect on stripe geometry implies skin effect on fully open-boundary geometry (i.e., the universal skin effect), which relies on a conjecture. The justification of this conjecture is discussed in the Supplementary Note 1.

The above theorem has implied the universality of skin effect in two and higher dimensions. As $E_i(\mathrm{BZ})$ is the image of the $d \geq 2$-dimensional torus on the complex plane, it takes fine tuning of parameters to make $A_i = 0$ for every $i$. In fact, for single-band Hamiltonian, we can prove that $A = 0$ if and only if $\mathcal{H}(\mathbf{k}) = P[h(\mathbf{k})]$, where $h(\mathbf{k})$ is a Hermitian Hamiltonian and $P$ is a complex polynomial (see Supplementary Note 1). In other words, a randomly generated non-Hermitian Hamiltonian $\mathcal{H}(\mathbf{k})$ has skin effect: the first meaning of universality. In previous studies, other types of skin effect, such as the line-skin and the high-order-skin effect, in two and higher dimensions have been proposed[30,34]. These types all require the open-boundary system take a special geometry (usually a rectangle) and are hence considered special and non-generic. Additionally, the number of skin modes in the universal skin effect follows a volume law, which differentiates from the higher-order-skin effect. The skin effect when $A_i \neq 0$ assumes a completely generic geometry of boundary: the second meaning of universality. The third meaning of universality lies in the fact that the higher-dimensional skin effect is compatible with all point-group symmetries, i.e., the universal skin effect can appear if and only if the spectral area is nonzero, regardless of the point-group symmetry of the bulk Hamiltonian. While in one dimension, if the bulk Hamiltonian only respects, for example, the inversion symmetry, the periodic-boundary energy spectrum has an

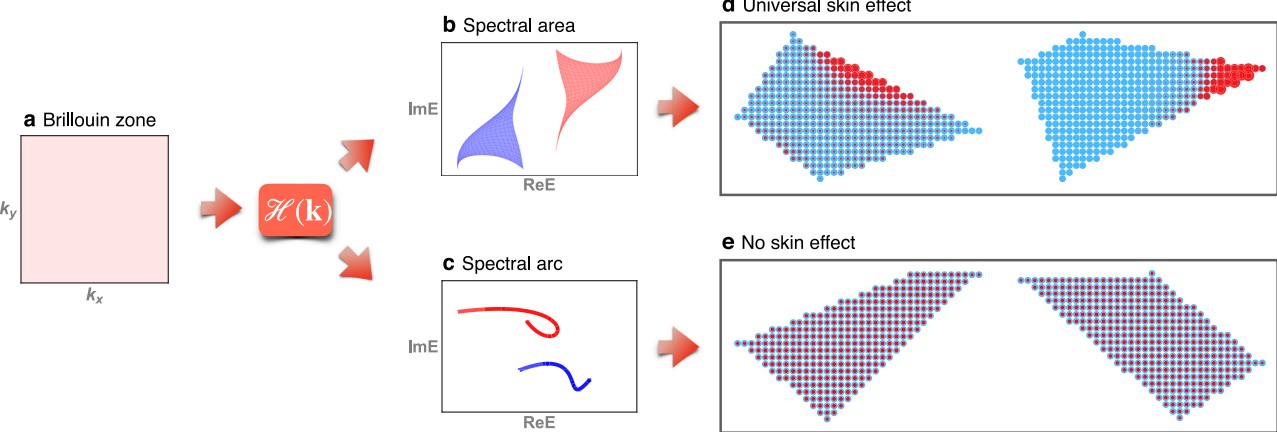

**Fig. 1 The theorem of universal skin effect. a** represents the Brillouin zone. **b, d** shows that if the spectral area of $\mathcal{H}(\mathbf{k})$ is nonzero, the skin effect will appear on some generic open-boundary geometries. **c, e** shows that when the spectral area of $\mathcal{H}(\mathbf{k})$ is zero, or forming one or several arcs on the complex plane, there is no skin effect under any geometry.

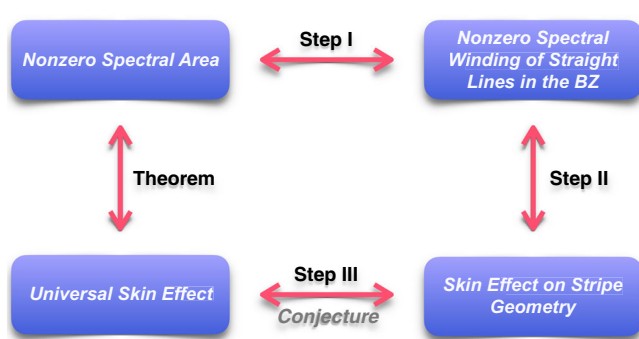

**Fig. 2 The sketch of the proof of the theorem.** The first step connects the nonzero spectral area and nonzero spectral winding number along some direction in the BZ. The latter means the Hamiltonian exhibits the skin effect under the corresponding stripe geometry, which is proved in the second step. The skin effect on stripe geometry further reveals the universal skin effect, relying on a conjecture in the third step.

arc-form on the complex plane, which means the absence of non-Hermitian skin effect[11,23]. A standing wave explanation for the theorem is provided in the Supplementary Note 2.

**The corner-skin and the geometry-dependent-skin effect.** While the theorem shows that the skin effect is universal, it does not specify what skin modes look like in higher dimensions. Here, we define the current functional, which partitions the universal skin effect into non-reciprocal skin effect and generalized reciprocal skin effect. Then we report the representative phenomena in these two types, i.e., the corner-skin effect (CSE) and the geometry-dependent skin effect (GDSE), respectively.

The current functional is defined as

$$J_{\boldsymbol{\alpha}}[n] = \sum_i \oint_{\mathrm{BZ}} dk^d \, n(E_i, E_i^*) \, \nabla_{\boldsymbol{\alpha}} E_i(\mathbf{k}) \qquad (1)$$

under the periodic-boundary condition, where $i$ is the energy band index and $\nabla_{\boldsymbol{\alpha}}$ indicates the directional derivative along certain direction $\boldsymbol{\alpha}$ in $d$-dimensional momentum space. Here, $n(E, E^*)$ represents a distribution function when the system reaches to a steady state and only depends explicitly on the energy of the system state[17]. The nonzero current functional (labeled by $J \neq 0$) is defined as: $\exists \, \boldsymbol{\alpha}, n, \, J_{\boldsymbol{\alpha}}[n] \neq 0$; and as a complementary set, the zero current functional (labeled by $J = 0$) is defined as: $\forall \, \boldsymbol{\alpha},$

$n, \, J_{\boldsymbol{\alpha}}[n] = 0$. By definition, the nonzero current functional and zero current functional are complete and mutually exclusive mathematically. Therefore, we can classify the universal skin effect (nonzero spectral area) into two types according to the current functional, i.e., the non-reciprocal skin effect ($J \neq 0$) and generalized reciprocal skin effect ($J = 0$), as illustrated in Fig. 3. Note that this classification of skin effect according to the current functional is different from the classification of intrinsic point-gap topology for symmetry class[11,18] (see Supplementary Note 3). The current functional is shown to vanish in two and three dimensions under point groups $C_i$, $D_{2,3,4,6}$, $C_{2h,3h,4h,6h}$, $D_{2d,3d,2h,3h,4h,6h}$, $T$, $T_{d,h}$, $O$ and $O_h$. Therefore, the non-reciprocal skin effect is only compatible with point groups $C_m$ and $C_{2,3,4,6,2v,3v,4v,6v}$. As a comparison, the generalized reciprocal skin effect is compatible with all point-group symmetries (see also Supplementary Note 3).

We define the CSE as a type of the non-reciprocal skin effect ($J \neq 0$) that exhibits the particular phenomenon that almost all eigenstates are localized at corners of the open-boundary geometry. The Hamiltonian of the example for CSE is

$$\mathcal{H}(\mathbf{k}) = [5(\cos k_x + \cos 2k_x) - i(\sin k_x + 3\sin 2k_x) + 5\cos k_y + i\sin k_y]/2, \qquad (2)$$

of which the spectral area under square geometry and triangle geometry is shown in Fig. 3(a, b) with light blue color. Because of the nonzero spectral area, the theorem tells us that the Hamiltonian must have the universal skin effect. This is verified in Fig. 3(c, d), where the spatial distributions of all eigenstates $W(\mathbf{x}) = \frac{1}{N}\sum_n |\psi_n(\mathbf{x})|^2$ under different open boundaries are plotted. Here $\psi_n(\mathbf{x})$ is a normalized right eigenstate and $N$ is the number of these eigenstates. It is found that the wave functions are always localized at the corner of the boundary in Fig. 3(c), even if the open-boundary geometry is changed in Fig. 3(d). We elaborate on the localization of eigenstates for this example in the Supplementary Note 4. We also plot the corresponding eigenvalue spectra under different open boundaries, as shown in Fig. 3(a, b) with red color. One can notice that the spectral areas under periodic and open boundaries do not equal. The CSE is a representative one of non-reciprocal skin effect and inherits its features, including nonzero current functional and incompatibility with certain point-group symmetries.

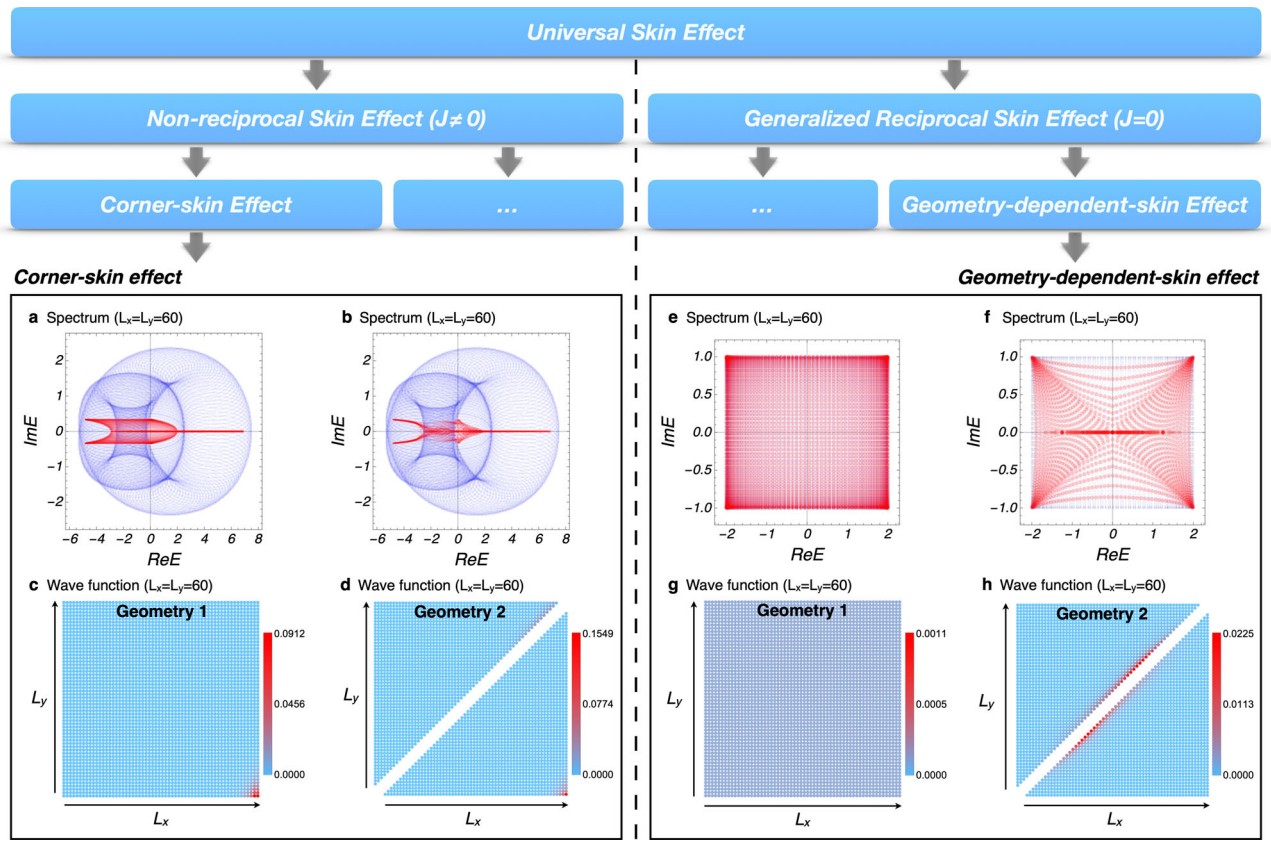

**Fig. 3 The corner-skin effect and the geometry-dependent-skin effect.** The universal skin effect can be further classified into two types by the current functional, that is, non-reciprocal skin effect ($\exists \alpha, n, J_\alpha[n] \neq 0$) and generalized reciprocal skin effect ($\forall \alpha, n, J_\alpha[n] = 0$). CSE (**a**)–(**d**) and GDSE (**e**)–(**h**) are representatives of these two types of skin effects, respectively. In (**a**, **b**, **e**, **f**), the light blue regions represent the spectrum under periodic boundary, where 200*200 **k**-grid is used, and the red points represent the eigenvalues under different open-boundary geometries. The system size under square geometry in (**c**, **g**) is $L_x \times L_y = 60 \times 60$, and each triangle geometry in (**d**, **h**) has the same right-angled side length $L_x = L_y = 60$. The spatial distributions of eigenstates $W(\mathbf{x})$ are plotted in (**c**, **d**, **g**, **h**) with the color bars. In the system with GDSE, the skin effect disappears under square geometry (geometry 1) in (**g**), and reappears under triangle geometry (geometry 2) in (**h**).

Similar to the definition of CSE, the GDSE is one type of generalized reciprocal skin effect ($J = 0$) showing the unique phenomenon that there is at least one fully open boundary geometry under which the skin effect does not appear. The Hamiltonian of the example for GDSE reads

$$\mathcal{H}(\mathbf{k}) = 2\cos k_x + i\cos k_y. \tag{3}$$

Since the spectral area is nonzero, our theorem tells us that the system must have skin effect for certain open-boundary geometry, such as a random polygon. However, an interesting phenomenon in this example is that the skin effect disappears under the square geometry due to the existence of two mirror symmetries shown in Fig. 3(g). Once we choose other types of boundaries where mirror symmetries are broken, the skin effect reappears as shown in Fig. 3(h). Since the appearance of the skin effect and the localization position depend on the geometry, it is called the GDSE. In one dimension, an open chain does not exhibit skin effect when its spectrum coincides with the corresponding periodic-boundary spectrum on the complex plane. Unlike in one dimension, even if the region covered by the energy spectrum under some open-boundary geometry (such as the triangle geometry in Fig. 3(h)) seems to be the same as the region covered by the periodic-boundary spectrum, the system can still show a skin effect due to the different density of states on the complex plane. It is also a unique feature in two- and higher-dimensional skin effects. In the Supplementary Note 4, we

provide some numerical results to illuminate this new type of skin effect and discuss the localization of eigenstates on the open-boundary geometry. In addition, we show that GDSE follows the volume law, i.e., the increase in the number of skin modes is proportional to the increase in the system volume. For GDSE, there is at least one spatial geometry such that skin modes vanish, and as such is mutually exclusive with CSE. Additionally, GDSE is compatible with all point groups, in contrast to CSE.

**Corollary: skin effect from exceptional points.** An immediate corollary of our theorem is that all lattice Hamiltonians with stable exceptional points have universal skin effect, connecting two unique phenomena in the non-Hermitian band theory. This connection has also been discussed in ref. [31], where the bands around the stable exceptional point form a point gap with non-zero spectral winding number, consequently, exhibiting the skin effect under an open-boundary geometry. Consider a stable exceptional point $\mathbf{k}_0$ in two dimensions. Due to the branch point structure of exceptional point, the dispersion around $\mathbf{k}_0$ can be expressed as[8] $E_\pm(\mathbf{k}) = \pm c_0 \sqrt{q_x + c_1 q_y} + O(|\mathbf{k} - \mathbf{k}_0|)$, where $q_{i=x,y}$ denotes a small derivation from exceptional point in $x$ or $y$ direction, that is, $q_i = k_i - k_{0i}$. Here $c_0, c_1$ are nonzero complex numbers and the stable exceptional point ensures the nonzero imaginary part of $c_1$. Suppose the range of the expansion is $r_0$, then it is clear that $A_\pm \geq |c_0|\pi r_0^2/2 \neq 0$. By the theorem, the

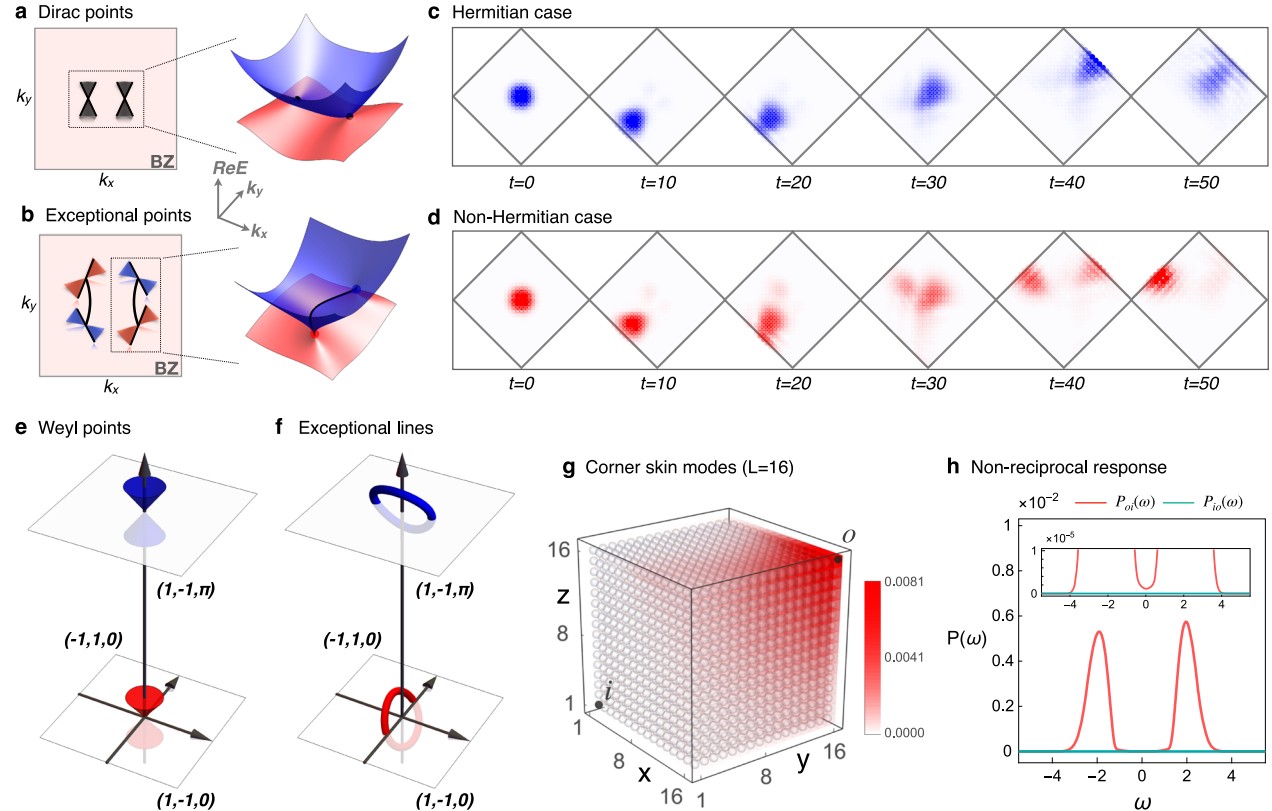

**Fig. 4 The photonic crystal model with exceptional points and Weyl semimetal with exceptional lines.** Two Dirac points (**a**) of a two-dimensional photonic crystal model are split into four exceptional points (**b**) upon adding non-Hermitian term, such as radiational loss. Correspondingly, the evolution of Gaussian wave packet with initial velocity at the center of a diamond geometry for each ten time intervals is shown in (**c**) with $\gamma = 0$ (Hermitian) and (**d**) with $\gamma = 1/4$ (GDSE). Two Weyl points (**e**) of a three-dimensional Weyl semimetal are expanded into two exceptional rings (**f**) after the addition of non-Hermitian perturbations. The spatial distribution of eigenstates, i.e. $W(\mathbf{x})$, is plotted in (**g**). The modulus square of the propagator from $i$ to $o P_{io}(\omega)$ and that from $o$ to $i P_{oi}(\omega)$, as functions of $\omega$, are plotted with red color and dark cyan color in (**h**), respectively.

system must have universal skin effect (see Supplementary Note 5).

Now we use the photonic crystal model that has been experimentally realized in ref. [45] to demonstrate our corollary. The tight-binding model Hamiltonian with periodic boundary can be written as

$$\mathcal{H}(\mathbf{k}) = \mathbf{d}(\mathbf{k}) \cdot \boldsymbol{\sigma} - i\gamma/2(\sigma_0 - \sigma_z), \quad (4)$$

where $\boldsymbol{\sigma} = (\sigma_0, \sigma_x, \sigma_y, \sigma_z)$ is a vector of the Pauli matrices and $\mathbf{d}(\mathbf{k})$ is a vector with four components, that is, $\mathbf{d}(\mathbf{k}) = \{\mu_0 - (t_2 + t_3)(\cos k_x + \cos k_y), t_1[1 - \cos k_x - \cos k_y + \cos(k_x - k_y)], t_1[\sin k_x - \sin k_y - \sin(k_x - k_y)], \mu_z + (t_2 - t_3)(\cos k_x - \cos k_y)\}$. The parameters are chosen as follows, $(t_1, t_2, t_3, \mu_0, \mu_z) = (0.4, -0.1, 0.5, 1.35, -0.02)$. As shown in Fig. 4(a), in the Hermitian limit, i.e. $\gamma = 0$, the system has two Dirac points along the $x$-axis. When external dissipation or radiational loss is added, i.e., $\gamma \neq 0$, each Dirac point splits into two exceptional points shown in Fig. 4(b), connected by the bulk Fermi arc. According to our theorem, the system must have the universal skin effect, more precisely, the GDSE. Specifically, the skin effect disappears under square geometry but reappears under diamond geometry, which is verified in the Supplementary Note 6.

So far, we have shown the features of the energy spectrum and wave function in the system with GDSE. We expect some observable phenomena from the skin effect, which motivates us to examine the dynamical properties for the photonic crystal model in Eq.(4). In order to show this, we simulate the time evolution of the wave packet starting at the center of the diamond geometry

with an initial velocity perpendicular to one edge. Here the initial state is chosen to be Gaussian form $|\psi_0\rangle = \mathcal{N} \exp[-(x - x_0)^2/10 - (y - y_0)^2/10 - i2x - i2y](1, 1)^T$, where $\mathcal{N}$ is the normalization factor and $x_0 = y_0 = 21$ is the center coordinate of the diamond geometry. We plot the corresponding spatial distribution of normalized final states $|\psi(t_f)\rangle = \mathcal{N}(t_f)e^{-i\mathcal{H}_{OBC}t_f}|\psi_0\rangle$ for every ten time intervals, where $\mathcal{H}_{OBC}$ represents the open-boundary Hamiltonian on the diamond geometry. As shown in Fig. 4(c), in the Hermitian case, the center of the wave packets obeys the simple law of reflection: the center of the wave packet just bounces between the two edges while slowly dispersing with time. However, in the non-Hermitian case ($\gamma = 1/4$) with GDSE, after several oscillations between two edges, the wave packet makes a side jump into the upper left corner as shown in Fig. 4(d). The transverse motion of the wave packet induced by skin effect is explained in more detail in the Supplementary Note 6. This anomalous dynamical behavior is an experimental signature of GDSE.

We also propose the realization for CSE in a three-dimensional system with exceptional lines. Consider a Weyl semimetal with non-Hermitian term as a perturbation, of which the periodic-boundary Hamiltonian reads

$$H(k) = [\mathbf{d}_r(\mathbf{k}) + i\delta \mathbf{d}_i(\mathbf{k})] \cdot \boldsymbol{\sigma}, \quad (5)$$

where $\mathbf{d}_r(\mathbf{k})$ and $\mathbf{d}_i(\mathbf{k})$ are vectors with four components, that is, $\mathbf{d}_r(\mathbf{k}) = (0, \sin k_x, \sin k_y, 2 - \cos k_x - \cos k_y + \sin k_z)$ and $\mathbf{d}_i(\mathbf{k}) = (-\sqrt{5}, 1 + \cos k_z, 1 - \cos k_z, \cos k_z)$. The Hermitian part $\mathbf{d}_r \cdot \boldsymbol{\sigma}$ is a Weyl semimetal possessing two Weyl points, the red cone at

$(0, 0, 0)$ and blue cone at $(0, 0, \pi)$ shown in Fig. 4(e). Upon turning on the non-Hermitian term, the Weyl points evolve into two exceptional rings in Fig. 4(f). Consequently, the system exhibits the CSE with $\delta = 1/6$ shown in Fig. 4(g), as a numerical verification of our corollary.

Experimentally, the non-reciprocity of the CSE can be detected by the two-point Green's function. The modulus square of the propagator from $i = (1, 1, 1)$ to $o = (16, 16, 16)$ is expressed as $P_{oi}(\omega) = \sum_{\alpha,\beta} |\langle o, \beta| \frac{1}{\omega - H} |i, \alpha\rangle|^2$, where $\alpha, \beta$ label the orbitals of the unit cell. We calculate $P_{oi}(\omega)$ and $P_{io}(\omega)$ in Fig. 4(h), where a significant difference between them demonstrates the non-reciprocity of CSE.

## Discussion

Our work has built a bridge between two distinct phenomena that only exist in non-Hermitian systems, i.e., the exceptional points (lines) and the non-Hermitian skin effect, by establishing the correspondence between bulk (spectral area) and boundary (universal skin effect). We prove that the skin effect is universal and compatible with all point-group symmetries and time-reversal symmetry in two and higher dimensions. Due to the universality, it is expected that the skin effect is observable in a wide range of platforms, such as photonic crystals with natural radiational loss, acoustic meta-materials and circuit networks with lossy components such as resistors. Beyond these classical systems, the skin effect can also be realized in condensed matter, e.g., the heavy-fermion material with finite quasiparticle lifetime and the Weyl-exceptional-ring semimetal. The latter is realizable in Weyl semimetals made from inverting bands that have disparate effective masses, such as d- and f-bands.

One should be reminded, however, that the results in this paper assume the coherent dynamics of the constituent degrees of freedom, which is unlikely the case in macroscopic condensed-matter systems where the coherence length is shorter than the system size. On the contrary, for the systems where the system size and the coherent length are comparable, as in mesoscopic systems, we believe that the universal skin effect has a significant contribution to the transport properties, a subject for future exploration.

## Data availability

Raw numerical data from the plots presented are available from the authors upon request.

## Code availability

The code used to generate the figures are available from the authors upon reasonable request.

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

## Acknowledgements

C.F. acknowledges funding support by the Ministry of Science and Technology of China under grant number 2016YFA0302400, and the Chinese Academy of Sciences under grant number XDB33000000. Z.Y. acknowledges funding support by the National Science Foundation of China (Grant No. NSFC-12104450) and the fellowship of China National Postdoctoral Program for Innovative Talents (Grant No. BX2021300).

## Author contributions

C.F. conceived the work; K.Z. did the major part of the theoretical derivation and numerical calculation; Z.Y. wrote and analyzed the tight-binding Hamiltonian of the photonic crystal model; All authors discussed the results and participated in the writing of the manuscript.

## Competing interests

The authors declare no competing interests.
