## [Peer Review File · Nature Communications]

REVIEWER COMMENTS

Reviewer #1 (Remarks to the Author):

In this work, the authors generalized the skin effect of non-Hermitian systems, which has been limited to 1D cases up to now, to two and higher dimensions. More specifically, it is shown that the skin effect in higher dimensional non-Hermitian systems exist, if and only if the energy spectrum under periodic boundary condition covers a finite area on the complex energy plane. Also, the authors proposed two types of skin effect, i.e., the corner skin effect (CSE) and the geometry-dependent skin effect (GDSE). As a corollary, it is shown that non-Hermitian systems with exceptional points (lines) in two (three) dimensions exhibit skin effect.

Considering the broad research interest of higher-dimensional non-Hermitian systems and the significance of the skin effect in non-Hermitian band theory, I think that the main results of the present manuscript can make a valuable contribution to the relevant research field. I find that the work potentially contains enough original results to warrant its publication in Nature Communications. However, although theoretical results shown in the paper are important and reliable, I find that there are several parts of the manuscript that should be improved further. Below I give several questions and comments which should be properly answered before I can make a final assessment on the paper.

(1) There are typos and grammatical errors in several sentences. For example, i) the first sentence of the third paragraph seems to be "Apart from" Instead of "Apart form". ii) in the sixth line of the fourth paragraph, "and "boundary" the presence (absence) of the skin effect", a verb should be added.

(2) It is stated that skin effect is compatible with spatial symmetry in higher dimensions while in one-dimension it is not. I think it is better to give more detailed discussion about this issue. Why is spatial symmetry incompatible in one-dimension?

(3) In Fig. 2(c), wave functions are shown to be localized at a corner. What determine the precise position (or which corner) where the states are localized?

(4) In Equation (2), $n(E, E^*)$ is explained to be an smooth function, whose meaning is unclear. Does its meaning depend on the type of current functional? I think it is better to define it more clearly.

(5) I think the description of the GDSE should be expanded more. What determine the position of localization in GDSE? Also, it is stated that "area of the open-boundary spectrum seems to be the same as A_i ". Can it be possible to make the argument more rigorous?

(6) Related to the skin effect from exceptional points, is it possible to have CSE from exceptional points, instead of GDSE? Also, the relation between the proposed theorem and the exceptional points is not so clear. Do exceptional points always support nonzero A_i ? If it is the case, it is better to show and mention it explicitly.

(7) In Fig.3(c,d), what determine the position where the eigenstates are localized? Also, I think it is good to expand Fig.3 and include the energy spectra under periodic and open boundary condition.

(8) There is a short description of the dynamical properties associated with the skin effect. Is it new? If it is the case, more detailed description is necessary. Why does the skin effect induce transverse motion of the wave packet?

(9) CSE in three-dimensional systems with exceptional lines should be properly described in the main text including relevant figures. Also, is it possible to observe GDSE for 3D systems with exceptional lines?

(10) It is stated that "skin effect is compatible with all spatial symmetries". But CSE seems to be compatible only with certain symmetries. The relevant statement should be properly modified.

Reviewer #2 (Remarks to the Author):

In the manuscript "Universal non-Hermitian skin effect in two and higher dimensions", the authors propose a universal theorem predicting the non-Hermitian skin effect in general systems solely from the spectral area covered by the complex eigenvalues in a periodic geometry. This general criterion provides a sought-after bulk-boundary correspondence and allows the distinction of two different flavours, the corner- and geometry-dependent skin effect. Following that, the authors relate the skin effect to the presence of exceptional points, potentially opening experimental verifications.

In the light of the multitude of works on higher order non-Hermitian skin effects which either relied on the presence of certain symmetries or applicability of topological invariants, this work stands out as providing a universal criterion for all of these systems. Moreover it differentiates purely geometrically from bulk induced skin effects and provides sound proofs as well as intuitive explanations for the presented statements. I find this an elegant work, and an important contribution to the field of non-Hermitian topology. I can thus recommend it for publication if the following points are clarified:

1) The authors present several eigenstate distribution plots, which lack colorbars. Especially when computed with a model presented in the manuscript, specific parameters (e.g. k-grid, system size, etc.) would also be useful in all figure captions to ensure reproducibility.

2) The authors present a corollary predicting a skin effect whenever stable exceptional points are present. This fact was already discussed in Ref.29 (Phys. Rev. Research 2, 023265), stating that the bands between two exceptional points necessarily form a point gap with non-trivial 1D winding number, therefore showing a skin effect under open boundaries. I think the authors should highlight this work, even though their approach does not rely on a topological invariant.

3) A feature of the GDSE seems to be that the area of the OBC spectrum is the same as in the PBC case. Can the authors comment on whether this is an artefact or feature, i.e. can one proof this relationship? Also I would suggest to complement Fig. 2 e,f with a density of states plot to supplement the discussion in the main text.

4) The authors provide a proof that the GDSE is absent in certain geometries. Is there a way to find the geometries not showing a skin effect without trying brute-force? This could be very useful in practical settings.

5) In order to highlight the influence of the skin effect on dynamical properties, the authors simulate the evolution of a wavepacket in a geometry with exceptional points. Since driven/dissipative systems are usually governed by the time-dependent Lindblad equations,

could the authors comment on why the Hermitian time evolution still holds in their setting?

Small remark: Please capitalize "Hamiltonian" throughout the manuscript.

Reviewer #3 (Remarks to the Author):

Review on "Universal non-Hermitian skin effect in two and higher dimensions" by Kai Zhang, Zhesen Yang, and Chen Fang.

First, I summarize what the authors study in this paper and new finding. In non-Hermitian systems, one of the important issues is the non-Hermitian skin effect (NHSE), where the spectrum strongly depends on the boundary conditions. For one-dimensional systems, it was shown in Refs. [17,18] that the nonzero winding number of complex spectra under the periodic boundary condition is a sufficient condition for the existence of the NHSE under the open boundary condition. In more general spatial dimensions, many examples of non-Hermitian Hamiltonians show a sort of NHSEs, and a systematic understanding of the NHSE in higher dimensions is required. In this paper, the authors propose a necessary and sufficient condition for the NHSE, described below. The central claim of this paper is that the NHSE occurs under any open boundary condition if and only if the complex spectrum has a finite area under the periodic boundary condition. Such a general condition for the NHSE in high dimensions has not yet been proposed except in this paper, and I think it is an impactful result.

While the results of this paper are significant and timely, the quality of the paper may not be up to the level of publication in the following points.

- 1) First of all, the main body of the paper says that it has proved a certain claim, but if we read the proof details of the supplemental material, it turns out that it depends partly on the following conjecture: When the winding number along a direction in the BZ is non-zero, the boundary perpendicular to that direction results in the NHSE. As the authors also pointed out in the Supplementary material, if all spatial directions are in open boundary conditions, and this is the situation of interest in the paper, this is still a conjecture. However, the main body of the paper does not explicitly state that part of the proof depends on the conjecture, which may cause misunderstanding among readers.
- 2) Related to the comment above, since this paper claims to have proved a certain theorem, I think the outline of the proof should be written in the main text.
- 3) The theorem claims that a finite area in the complex spectrum causes the NHSE in *any* open boundary condition. However, GDSE says that there is at least one geometry where the NHSE disappears, which contradicts the claim of the theorem. Therefore, the statement of the theorem should be refined so that it is consistent with GDSE.
- 4) For the CSE, there are only examples and no definition of CSE. Sec. IIIB of the Supplemental material shows that it is equivalent to the current functional being non-zero, but how do we define it as a phenomenon?
- 5) Also, there is no definition for GDSE, only examples.
- 6) Regarding (ii) of "Universality," while the universal NHSE is claimed to be compatible with spatial and time-reversal symmetries, the only symmetry discussed in Supplemental Material is unitary point group symmetry and there is no discussion of the case where time-reversal symmetry is included. Also, what kind of time-reversal symmetry should we

consider? For example, in [11], a "time-reversal symmetry" as the transpose is introduced. Also, are there any restrictions on other kinds of internal symmetries (e.g., pseudo Hermiticity)?

7) Also related to "Universality" (ii), the discussion in Supplemental Material III-B-2 seems to be limited to three spatial dimensions. How do we understand that two-fold rotation and inversion symmetry are the same symmetry in two dimensions (except for the factor system), but the only two-fold rotation is compatible with universal NHSE?

I also have several minor comments listed below.

8) I think it is worth stating that the open boundary condition in a two-dimensional system in the claim of the theorem is the full open boundary condition for all directions.

9) Regarding NHSE of interest in the paper, I would suggest commenting somewhere that the number of skin modes follows a volume law, for example, to clarify the difference from the higher-order skin effects.

10) Regarding Fig. 2 d, the upper and lower sites of the geometry are considered to be identical, so I think the "triangle geometry" may be misleading.

11) On the left part of page 4, is the estimation for A_{\pm} correct? If the imaginary part of c_1 is infinitesimal, then the area A_{\pm} is also infinitesimal.

12) On the right part of page 4, it is written, "As mentioned in the previous discussion, the appearance of the skin effect can be reflected in the dynamical properties." However, the relationship between dynamical properties of the system and the skin effect is not mentioned anywhere in the previous section.

13) "Lei algebra"

II. RESPONSE TO REVIEWER #1

“In this work, the authors generalized the skin effect of non-Hermitian systems, which has been limited to 1D cases up to now, to two and higher dimensions. More specifically, it is shown that the skin effect in higher dimensional non-Hermitian systems exist, if and only if the energy spectrum under periodic boundary condition covers a finite area on the complex energy plane. Also, the authors proposed two types of skin effect, i.e., the corner skin effect (CSE) and the geometry-dependent skin effect (GDSE). As a corollary, it is shown that non-Hermitian systems with exceptional points (lines) in two (three) dimensions exhibit skin effect.

Considering the broad research interest of higher-dimensional non-Hermitian systems and the significance of the skin effect in non-Hermitian band theory, I think that the main results of the present manuscript can make a valuable contribution to the relevant research field. I find that the work potentially contains enough original results to warrant its publication in Nature Communications. However, although theoretical results shown in the paper are important and reliable, I find that there are several parts of the manuscript that should be improved further. Below I give several questions and comments which should be properly answered before I can make a final assessment on the paper.”

We are grateful for the reviewer’s recognition of the importance and novelty of our work, as he/she writes: “I think that the main results of the present manuscript can make a valuable contribution to the relevant research field. I find that the work potentially contains enough original results to warrant its publication in Nature Communications.”.

We have comprehensively modified our manuscript according to the reviewer’s suggestions, and the revised details are below the corresponding questions. We thank the reviewer again for the valuable comments, which greatly improve the quality and completeness of our manuscript.

“(1) There are typos and grammatical errors in several sentences. For example, i) the first sentence of the third paragraph seems to be “Apart from ...” Instead of “Apart form”. ii) in the sixth line of the fourth paragraph, “and “boundary” the presence (absence) of the skin effect”, a verb should be added.”

We thank our reviewer for his/her careful reading of our manuscript.

In the revised version, we have rechecked and corrected the typos and grammatical errors. Especially, according to the second suggestion of the reviewer,

“and boundary the presence (absence) of the skin effect”
has been changed to

“and boundary means the presence (absence) of the skin effect”.

“(2) It is stated that skin effect is compatible with spatial symmetry in higher dimensions while in one-dimension it is not. I think it is better to give more detailed discussion about this issue. Why is spatial symmetry incompatible in one-dimension?”

We thank our reviewer for pointing out this interesting and important question.

Next, we explain why is spatial symmetry incompatible in one dimension but compatible in higher dimensions.

In one dimension, a sufficient and necessary condition for the absence of skin effect is the vanishing of the spectral winding of $\mathcal{H}(k)$ with respect to any given base energy $E_b \in \mathbb{C}$ [1, 3], that is

$$\forall E_b \in \mathbb{C}, \quad w_{E_b} = \frac{1}{2\pi i} \int_0^{2\pi} d \ln \det[\mathcal{H}(k) - E_b] = 0. \quad (\text{R1})$$

This zero spectral winding condition is equivalent to Bloch spectrum being one or several arcs on the complex energy plane. It can be shown when the Hamiltonian $\mathcal{H}(k)$ has inversion symmetry, the above spectral winding number must be zero. As a result, it is impossible to have the skin effect. Let’s take an example to illustrate this point. Consider a single-band model with

inversion symmetry, the Hamiltonian is invariant when k is reversed, i.e., $\mathcal{H}(k) = \mathcal{H}(-k)$. Then,

$$\begin{aligned} w_{E_b} &= \frac{1}{2\pi} \int_{-\pi}^{\pi} \partial_k \arg[\mathcal{H}(k) - E_b] dk = \frac{1}{2\pi} \int_{-\pi}^{\pi} \partial_{-k} \arg[\mathcal{H}(-k) - E_b] dk \\ &= \frac{1}{2\pi} \int_{-\pi}^{\pi} \partial_{-k} \arg[\mathcal{H}(k) - E_b] dk = -w_{E_b} = 0 \end{aligned} \quad (\text{R2})$$

for all $E_c \in \mathbb{C}$. The above argument indicates that the skin effect is incompatible with point groups including inversion symmetry in one dimension. For other symmetries, please refer to the previous works [4, 5].

However, in higher dimensions, as discussed in the main text, the sufficient and necessary condition for the absence of skin effect under any open-boundary geometry is the vanishing of the spectral area, and no point-group symmetry can guarantee this condition.

For example, consider a two-dimensional single-band Hamiltonian $\mathcal{H}(\mathbf{k}) = 2 \cos k_x + i \cos k_y$ as an example. Although this Hamiltonian respects two mirror symmetries $\mathcal{M}_x, \mathcal{M}_y$ and their combination—inversion symmetry, its spectral area is nonzero. To be more specific, although the mirror- x symmetry, $\mathcal{M}_x \mathcal{H}(k_x, k_y) \mathcal{M}_x^{-1} = \mathcal{H}(-k_x, k_y)$, restricts the spectrum of the one-dimensional k_y -subsystem to be an arc-form, these arcs for different k_y can stack to form a nonzero spectral area on the complex energy plane. Therefore, there is no such point-group symmetry under which the spectral area must be zero. As shown in Fig. 3(g)(h) of the main text, when the boundary condition preserves/breaks the two mirror symmetries, the skin effect disappears/reappears (an explanation can be found in the reply of the fourth question from reviewer #2). The numerical results show that the skin effect is compatible with point-group symmetry in higher dimensions.

The above discussion explains why we state that spatial symmetry is incompatible with skin effect in one-dimension but compatible in higher dimensions. The compatibility with all point-group symmetries is one of the meanings of “universality” of the higher-dimensional skin effect.

 Due to the well discussion of the relation between symmetries and one-dimensional skin effect in the previous works, in the revised version, we have made the following corrections.

1. In order to make the statement more precisely, “spatial symmetry” has been replaced by “point-group symmetry” throughout the revised manuscript.
2. In the main text, we have replaced the penultimate sentence of the “Theorem” section by the following one

“The third meaning of universality lies in the fact that the higher-dimensional skin effect is compatible with all point-group symmetries, i.e., the universal skin effect can appear if and only if the spectral area is nonzero, regardless of the point-group symmetry of the bulk Hamiltonian. While in one dimension, if the bulk Hamiltonian respects, for example, the only inversion symmetry, the periodic-boundary energy spectrum has an arc-form on the complex plane, which means the absence of non-Hermitian skin effect [4, 5].”

Here Ref. [4, 5] are added, which will guide the readers to the one-dimensional cases.

“(3) In Fig. 2(c), wave functions are shown to be localized at a corner. What determine the precise position (or which corner) where the states are localized?”

We are grateful to our reviewer for this valuable problem. The precise position where the eigenstates are localized is determined by the generalized Brillouin zone (GBZ). As will be explained in the following contents, in the model discussed in Fig. 3(c) (that is Fig. 2(c) in the original version), the GBZ can be calculated exactly. However, for a generic model with arbitrary open boundary condition, the calculation of the corresponding GBZ is still an open question.

 Now we calculate the GBZ of the model discussed in Fig. 3(c).

We first write down the Hamiltonian discussed in Fig. 3(c).

$$\mathcal{H}(\mathbf{k}) = [5(\cos k_x + \cos 2k_x) - i(\sin k_x + 3 \sin 2k_x) + 5 \cos k_y + i \sin k_y]/2. \quad (\text{R3})$$

Notice that under the open boundary condition shown in Fig. 3(c), the system can be solved by the separation of variables method. By replacing e^{ik_x} and e^{ik_y} with complex variables β_x and β_y , respectively, the Hamiltonian can be rewritten as a

FIG. R1. The two-dimensional GBZ of Hamiltonian Eq. (R4) is the direct product of y -subsystem GBZ [the red circle in (a)] and x -subsystem GBZ [the blue loop in (b)], and for comparison, the Brillouin zone is plotted as the dashed gray unit circle. The 2D GBZ reproduces the open-boundary spectrum under the thermodynamic limit, shown as the light blue continuum in (c), which is consistent with the numerically calculated eigenvalues (the red points) of the Hamiltonian on a square geometry with system size $L_x = L_y = 50$. In addition, the spatial distribution of eigenstates is shown in (d).

polynomial

$$\mathcal{H}(\beta_x, \beta_y) = h_x(\beta_x) + h_y(\beta_y), \quad (\text{R4})$$

where

$$h_x(\beta_x) = \frac{\beta_x^2}{2} + \beta_x + \frac{3}{2\beta_x} + \frac{2}{\beta_x^2}, \quad h_y(\beta_y) = \frac{3\beta_y}{2} + \frac{1}{\beta_y}. \quad (\text{R5})$$

For the x - and y - directions, the corresponding 1D GBZs are determined by the following characteristic equations

$$h_x(\beta_x) - E_x = 0, \quad h_y(\beta_y) - E_y = 0, \quad (\text{R6})$$

respectively. Using the approach developed in previous Refs. [6–8], the results are shown in Fig. R1 (a) and (b), where \mathbb{T}_x and \mathbb{T}_y denote the GBZs in the x - and y - directions, respectively. Having obtained \mathbb{T}_x and \mathbb{T}_y , the corresponding asymptotic energy spectrum of the 2D system in the thermodynamic limit becomes

$$E_{\text{OBC}} = h_x(\beta_x \in \mathbb{T}_x) + h_y(\beta_y \in \mathbb{T}_y), \quad (\text{R7})$$

which is shown as the light blue region in Fig. R1 (c).

Now the precise localization position of the eigenstates can be analyzed the corresponding 2D GBZ, which is $\mathbb{T}_x \times \mathbb{T}_y$ in this model. Since the radius of \mathbb{T}_y is less than 1, all eigenstates are localized in the negative y -direction. In addition, the radius of \mathbb{T}_x greater than 1 indicates that all eigenstates concentrate in the positive x -direction. Therefore, the open-boundary eigenstates are localized in the lower right corner, as shown in Fig. R1(d).

We finally note that when the boundary condition breaks the separation of variables condition, there is no standard method to calculate the GBZ.

In the revised version, we have added the above calculation and discussion in the Supplementary Note 4-A-2.

“(4) In Equation (2), $n(E, E^*)$ is explained to be a smooth function, whose meaning is unclear. Does its meaning depend on the type of current functional? I think it is better to define it more clearly.”

We thank our reviewer for the important question and helpful suggestion, which motivates us to unearth the meaning of the current function more deeply. In the revision, we have made a systematic modification about the current functional and higher-dimensional skin effect, which adequately improves the quality of our manuscript.

$n(E, E^*)$, which is assumed that only depends explicitly on the complex energy of system states, represents the distribution function when the system reaches to a non-equilibrium steady state. In addition, the current functional $J_\alpha[n]$ is defined as the function of the distribution function $n(E, E^*)$. Therefore, different input $n(E, E^*)$, which corresponds to different non-equilibrium steady state, gives different output $J_\alpha[n]$ with fixed α .

In what follows, we first give a clear definition of the current functional, including “nonzero current functional” and “zero current functional”. Then, we classify the two- and higher-dimensional skin effect into two types by the current functional, which shows one advantage of the current functional.

1. The definition of the current functional

In d dimensions, generally, the current functional is defined as

$$J_{\alpha}[n] = \sum_{\mu} \oint_{\text{BZ}} dk^d n(E_{\mu}, E_{\mu}^*) \nabla_{\alpha} E_{\mu}(\mathbf{k}) \quad (\text{R8})$$

under the periodic-boundary condition, where μ represents the energy band index and ∇_{α} is the directional derivative along certain direction $\alpha = \sum_{i=1}^d \alpha_i \hat{e}_{k_i}$ in d -dimensional momentum space (Here \hat{e}_{k_i} represents the i -th basis in momentum space). Here, $n(E, E^*)$ represents a distribution function when the system is in a steady state and only depends explicitly on the complex energy of the system state. Therefore, the current functional $J_{\alpha}[n]$ is defined as the function of the distribution function $n(E, E^*)$, and different input $n(E, E^*)$ gives different output $J_{\alpha}[n]$ with fixed α . In Hermitian case, the current functional becomes $J_{\alpha}[n] = \sum_{\mu} \oint_{\text{BZ}} dk^d n(E_{\mu}) \nabla_{\alpha} E_{\mu}(\mathbf{k}) = \sum_{\mu} \oint_{\text{BZ}} dk^d n(E_{\mu}) v_{\mu, \alpha}(\mathbf{k}) = 0$, where $v_{\mu, \alpha}(\mathbf{k}) = \sum_{i=1}^d \alpha_i \frac{\partial E_{\mu}(\mathbf{k})}{\partial k_i}$ is the group velocity along direction α in the μ -th energy band. In non-Hermitian systems, the energy E is generally a complex number, therefore, the directional derivative of $E_{\mu}(\mathbf{k})$ along α in Eq.(R8) becomes

$$\nabla_{\alpha} E_{\mu}(\mathbf{k}) = \sum_{j=1}^d \alpha_j \left(\frac{\partial \text{Re} E_{\mu}(\mathbf{k})}{\partial k_j} + i \frac{\partial \text{Im} E_{\mu}(\mathbf{k})}{\partial k_j} \right), \quad (\text{R9})$$

which represents the generalized (complex) velocity in d dimensions.

Specially, in one-dimensional non-Hermitian system, the current functional reduces to

$$J^{1\text{D}}[n] = \sum_{\mu} J_{\mu}^{1\text{D}}[n] = \sum_{\mu} \oint_{\text{BZ}} dk n(E_{\mu}, E_{\mu}^*) \partial_k E_{\mu}(k), \quad (\text{R10})$$

where μ represents the band index. The generalized velocity for μ -th band Eq.(R9) becomes

$$\partial_k E_{\mu}(k) = \partial_k \text{Re} E_{\mu}(k) + i \partial_k \text{Im} E_{\mu}(k), \quad (\text{R11})$$

which corresponds to the tangent vector of the μ -th energy band on the complex plane.

Based on the above definition of current functional, we further define the “nonzero current functional” as,

$$\exists \alpha, n; J_{\alpha}[n] \neq 0, \quad (\text{R12})$$

which is simply labeled as $J \neq 0$. As the complementary set, the “zero current functional” (labeled by $J = 0$) is defined as

$$\forall \alpha, n; J_{\alpha}[n] = 0, \quad (\text{R13})$$

which means the current functional is zero regardless of the choice of α and $n(E, E^*)$. For example, a Hermitian system always has zero current functional [1].

2. The classification of skin effect by the current functional

The nonzero current functional and zero current functional together constitute a complete set mathematically, and they are mutually exclusive. Based on this, we can completely classify the skin effect according to the zero and nonzero current functional as shown in Fig. R2.

- We first review of the one-dimensional skin effect.

In one dimension, the current functional reduces to Eq.(R10). In Ref. [1], the authors claim that for a one-dimensional system without any symmetry, if it has nonzero current functional ($J \neq 0$) in Eq.(R12), then the system has skin effect (which

FIG. R2. The universal skin effect can be further classified into two types by the current functional, that is, non-reciprocal skin effect ($\exists \alpha, n; J_\alpha[n] \neq 0$) and generalized reciprocal skin effect ($\forall \alpha, n; J_\alpha[n] = 0$). CSE and GDSE are representative one in these two types skin effect, respectively.

is called \mathbb{Z} skin effect), and vice versa. Another type of 1D skin effect — \mathbb{Z}_2 skin effect has been reported in Ref. [3]. In this case, the system need to respect the spinful anomalous time-reversal symmetry, namely, $\mathcal{U}_T^\dagger \mathcal{H}^t(k) \mathcal{U}_T = \mathcal{H}(-k)$ with $\mathcal{U}_T \mathcal{U}_T^\dagger = -1$. In fact, we can prove that for the \mathbb{Z}_2 skin effect, the system always has zero current functional ($J = 0$) in Eq.(R13). A simple proof is present as follows.

This anomalous time-reversal symmetry requires the energy bands always come in pair and satisfy $E_\uparrow(k) = E_\downarrow(-k)$. Correspondingly, for this pair of energy bands, the current functional satisfies $J_\uparrow[n] = -J_\downarrow[n]$ (See more details in the Supplementary Note 3-C-2). Here $n(E, E^*)$ is invariant under the anomalous time-reversal symmetry, because n only depends on the energy and the complex energy is invariant under this symmetry. Therefore, the current functional for all energy bands Eq.(R10) must be sum up to zero, regardless of the distribution function $n(E, E^*)$.

To sum up, the 1D skin effect can be completely classified into \mathbb{Z} skin effect with nonzero current functional ($J \neq 0$) and \mathbb{Z}_2 skin effect with zero current functional ($J = 0$). (Note that this classification of skin effect according to the current functional is different from the classification of intrinsic point-gap topology for symmetry class in Ref. [3].)

- Then we classify the universal skin effect into two types by the current functional.

The theorem tells us that in two and higher dimensions the system has the universal skin effect, if and only if the spectral area is nonzero. According to the current functional, the universal skin effect can be further classified into non-reciprocal skin effect with nonzero current functional ($J \neq 0$) and generalized reciprocal skin effect with zero current functional ($J = 0$), as shown in Fig. R2. By definition in Eq.(R12) and Eq.(R13), these two types of skin effect are complete and mutually exclusive.

Please see more detailed descriptions about non-reciprocal skin effect and generalized reciprocal skin effect in the Supplementary Section III-B-2.

Here we emphasize that the current functional provide a general framework to further examine the higher-dimensional skin effect. We can see that these two types of skin effect has essentially different appearance. To demonstrate this, we further specify two representative phenomena in these two types of skin effect, that is, the corner-skin effect (CSE) and geometry-dependent-skin effect (GDSE) shown in Fig. R2. More descriptions about CSE and GDSE have been discussed in the Section “The corner-skin and the geometry-dependent-skin effect” of the main text.

The above discussed classification of skin effect is one advantage of the current functional. Another advantage is that we can analyze the restriction of all point-group symmetries on the current functional, and further obtain the compatibility or incompatibility of the skin effect with all point groups. This analysis about all point groups have been completed in the Supplementary Note 3-C-1.

 In the revised version, we have made the following modifications.

1. In the main text, we have added more description about the current functional in the section “The corner-skin and the geometry-dependent-skin effect”.

2. We have added above discussion and more details into the Supplementary Note 3.

“(5.1) I think the description of the GDSE should be expanded more. What determine the position of localization in GDSE?”

As replied in question (3), the precise localization position of the wavefunctions in the GDSE is fully determined by the higher-dimensional GBZ theory, which is difficult to obtain in general case. Nevertheless, here we can provide a method to determine which edge (or surface) the eigenstates are localized on for a given open-boundary geometry.

Now we explain how to obtain the localization boundary. As shown in Fig. R3, our strategy is to define a winding number for each edge (red one in Fig. R3), and using this winding number to characterize the localization properties of the eigenstates.

Now we define the winding number. In two dimensions, for each edge of the open-boundary geometry, we first transform the momentum basis from $\mathbf{k} = (k_x, k_y)^T$ to $\mathbf{q} = (q_{\parallel}, q_y)^T$ by

$$\mathbf{k} = S\mathbf{q}; \quad \det S = 1. \quad (\text{R14})$$

Here, q_{\parallel} is chosen as the momentum parallel to the edge, and S is a 2×2 matrix with unitary determinant, which is the element of group $\text{SL}_2(\mathbb{Z})$. Note that the choice of q_y is not unique, it only needs to satisfy Eq.(R14). Accordingly, the Hamiltonian transforms from $\mathcal{H}(k_x, k_y)$ to $\tilde{\mathcal{H}}(q_{\parallel}, q_y)$. Based on the above definition, we state that if the following condition is satisfied,

$$\forall q_{\parallel} \in [-\pi, \pi], \quad \forall E_b \in \mathbb{C}, \quad w_{E_b}(q_{\parallel}) = \frac{1}{2\pi i} \int_{-\pi}^{\pi} dq_y \partial_{q_y} \log \det[\tilde{\mathcal{H}}(q_{\parallel}, q_y) - E_b] = 0, \quad (\text{R15})$$

then, no eigenstates are localized at this edge parallel to q_{\parallel} . On the contrary, if the above condition is destroyed, some eigenstates must be localized at the corresponding edge. Note that this statement is strictly true when the periodic boundary is adopted in the other directions, and it is our conjecture when the open boundary is applied in other directions. We believe this conjecture is true from many numerical supports and physical arguments but cannot prove it rigorously. For more details, please refer to the answer to the question (1) from reviewer #3.

Now we take the single-band example of GDSE to explain it in more detail.

The bulk Hamiltonian reads

$$\mathcal{H}(\mathbf{k}) = 2 \cos k_x + i \cos k_y. \quad (\text{R16})$$

For the “edge 1” and “edge 2” of the triangle geometry in Fig. R3(a)(c), the transformation matrix S equal to $\{\{1, 0\}, \{0, 1\}\}$ and $\{\{1, -1\}, \{0, 1\}\}$, respectively. Obviously, for any fixed k_x , the spectral winding number $w_{E_b}(k_x) = 0$, as shown in Fig. R3(b). The zero spectral winding number is guaranteed by the mirror symmetry in y direction of the bulk Hamiltonian, $\mathcal{H}(k_x, k_y) = \mathcal{H}(k_x, -k_y)$. For “edge 2” of the geometry, the Hamiltonian can be transformed into

$$\tilde{\mathcal{H}}(\mathbf{q}) = 2 \cos(q_{\parallel} - q_y) + i \cos q_y. \quad (\text{R17})$$

For the fixed q_{\parallel} , the spectrum of $\tilde{\mathcal{H}}(\mathbf{q})$ forms an closed loop on the complex plane, as shown in Fig. R3(d). The above process can be taken for every edge of the open-boundary geometry. Therefore, the eigenstates are localized on the “edge 2” instead of “edge 1” of the triangle geometry, as shown in Fig. 3(h) of the main text.

FIG. R3. To probe the skin effect on edge 1 and edge 2 of the triangle geometry, one needs to reselect the corresponding momentum basis as shown in (a) and (c), respectively. The spectra of the Hamiltonian Eq.(R17) in E - k_x and E - q_{\parallel} space are plotted in (b) and (d), respectively, where the blue region represents the spectral projection onto the complex energy plane.

According to the reviewer’s suggestion, in our revised version, we have explained GDSE in more detail in the last paragraph of the section “The corner-skin and the geometry-dependent-skin effect”, and added the above discussion into Supplementary Note 4-B-2.

“(5.2) Also, it is stated that “area of the open-boundary spectrum seems to be the same as A_i ”. Can it be possible to make the argument more rigorous?”

We thank our reviewer for pointing out this issue. Strictly speaking, we cannot prove this statement exactly. It comes from our numerical observation.

Here we show some numerical results to support this statement.

FIG. R4. The number of the unit cell in x and y directions L of the four triangle geometries (a-d) are taken as 20, 40, 60, 80, respectively. The spatial distribution of the eigenstates of the Hamiltonian Eq.(R16) on these triangle geometries are shown in (a-d), and the corresponding spectra on the complex plane are shown in (e-h), where the light blue region represents the spectral area A .

We calculate the eigenstates and spectra of Hamiltonian Eq.(R16) on the triangle geometry with different system size. From Fig. R4 we can observe that as the system size increases, the area covered by the open-boundary spectrum on the complex plane becomes gradually larger. It can be expected that the area of the open-boundary spectrum under the thermodynamic limit tend to be the same as the spectral area (the light blue region in Fig. R4).

We finally note that even if the area of the energy spectrum under some open-boundary geometry seems to be the same as the spectral area A_i , the system still has a skin effect due to the different density of states on the complex-energy plane (the density of states plot has been supplemented in question (3.2) from the Reviewer #2).

In the revised version, the following revision has been made:

- We have adjusted the statement

“Besides the distribution of the eigenstates, another feature of the GDSE is that area of the open-boundary spectrum seems to be the same as A_i . However, the corresponding density of states is dependent by the choice of geometry as shown in Fig. 2 (e)(f). We conjecture this is a universal phenomenon for the GDSE.”

in the last paragraph of the section “The corner-skin and the geometry-dependent-skin effect” as

“In one dimension, an open chain does not exhibit skin effect when its spectrum coincides with the corresponding periodic-boundary spectrum on the complex plane. Unlike in one dimension, even if the region covered by the energy spectrum

under some open-boundary geometry (such as the triangle geometry in Fig. 3(h)) seems to be the same as the region covered by the periodic-boundary spectrum, the system can still show a skin effect due to the different density of states on the complex plane. It is also a unique feature in two- and higher-dimensional skin effects.”

- The above statements have been added into the Supplementary Note 4-B-3.

“(6.1) Related to the skin effect from exceptional points, is it possible to have CSE from exceptional points, instead of GDSE?”

FIG. R5. (a) shows the real-part energy band structure with two exceptional points located at $(k_x, k_y) \approx (\mp 0.498, \pm 0.498)$. The spatial distribution of eigenstates of the Hamiltonian Eq. (R18) with open boundaries is plotted in (b), where the system size is $L_x = L_y = 30$.

The answer to this question is yes.

Now we show an example to illustrate this point.

Consider the following Hamiltonian

$$\mathcal{H}(k_x, k_y) = (\sin k_x + i)\sigma_x + (\sin k_y + i)\sigma_y + (3 - \cos k_x - \cos k_y)\sigma_z - \sqrt{5}i. \quad (\text{R18})$$

The real-part energy band structure is shown in Fig. R5(a), where two exceptional points connected by an bulk Fermi arc (the white line) lie in the $k_x - k_y$ direction. If we open boundaries along x and y directions, the corner-skin effect will appear in the upper right corner, as shown in Fig. R5(b). Therefore, it is possible to have CSE from exceptional points.

“(6.2) Also, the relation between the proposed theorem and the exceptional points is not so clear. Do exceptional points always support nonzero A_i ? If it is the case, it is better to show and mention it explicitly.”

The corollary of our theorem discussed in the main text is that

“all lattice Hamiltonians having stable exceptional points have universal skin effect”.

Here the “stable” exceptional point refers to the exceptional point whose topological charge (discriminant number) is ± 1 [9]. For the stable exceptional point, the spectral area A_i must be nonzero. This fact is guaranteed by their topological properties.

Now, we will demonstrate it.

In order to simplify the discussion, we here only consider a two-band model, whose Bloch Hamiltonian can be written as

$$\mathcal{H}(\mathbf{k}) = \sum_{i=x,y,z} h_i(\mathbf{k})\sigma_i, \quad (\text{R19})$$

where $h_i(\mathbf{k}) = h_i^r(\mathbf{k}) + ih_i^i(\mathbf{k})$. Here we have omitted the $h_0(\mathbf{k})\sigma_0$ term, which is irrelevant to the discussion of exceptional points. The eigenvalues of the Hamiltonian is

$$E_{\pm}(\mathbf{k}) = \pm\sqrt{\Delta(\mathbf{k})} = \pm\sqrt{h_x^2(\mathbf{k}) + h_y^2(\mathbf{k}) + h_z^2(\mathbf{k})}. \quad (\text{R20})$$

The emergence of the exceptional point \mathbf{k}_{EP} requires that

$$\Delta(\mathbf{k}_{\text{EP}}) = h_x^2(\mathbf{k}_{\text{EP}}) + h_y^2(\mathbf{k}_{\text{EP}}) + h_z^2(\mathbf{k}_{\text{EP}}) = 0 \quad (\text{R21})$$

and there exists an invertible matrix P such that

$$P^{-1}\mathcal{H}(\mathbf{k}_{\text{EP}})P = \begin{pmatrix} 0 & a \\ 0 & 0 \end{pmatrix} \quad (\text{R22})$$

with $a \neq 0$ a general complex number. The topological charge of the exceptional point is

$$\nu(\mathbf{k}_{\text{EP}}) = \frac{1}{2\pi i} \oint_{\Gamma(\mathbf{k}_{\text{EP}})} d\mathbf{k} \cdot \nabla_{\mathbf{k}} \ln \Delta(\mathbf{k}). \quad (\text{R23})$$

which describes the winding number of the Bloch spectrum around $E_0 = E_{\pm}(\mathbf{k}_{\text{EP}}) = 0$ when \mathbf{k} winds around the \mathbf{k}_{EP} . Thus one can imagine that when $\nu(\mathbf{k}_{\text{EP}}) \neq 0$, the corresponding spectral area must be nonzero. Here we note that the above derivation does not require that the degeneracy point is an exceptional point. Actually, any degeneracy point with nonzero topological charge means the nonzero spectral area.

Now we consider the two-dimensional system with stable exceptional points (that is $\nu(\mathbf{k}_{\text{EP}}) = \pm 1$). The dispersion around the exceptional point \mathbf{k}_{EP} can be expanded [10] as

$$E_{\pm}(\mathbf{q}) = \pm \sqrt{c_x q_x + c_y q_y} + O(|\mathbf{q}|), \quad (\text{R24})$$

where c_x, c_y are nonzero complex numbers and $c_{x/y} = c_{x/y}^r + i c_{x/y}^i$ with the superscript r/i indicates the real/imaginary part. The above equation implies that

$$\Delta(\mathbf{q}) \simeq c_x q_x + c_y q_y = c_x \tilde{\Delta}(\mathbf{q}) = c_x (q_x + c_1 q_y) \quad (\text{R25})$$

with $c_1 = c_y/c_x = c_1^r + i c_1^i$, where $\tilde{\Delta}(\mathbf{q}) = \text{Re} \tilde{\Delta}(\mathbf{q}) + i \text{Im} \tilde{\Delta}(\mathbf{q}) = (q_x + c_1^r q_y) + i c_1^i q_y$. Putting this equation into the topological charge formula Eq.(R11), equivalently, one can obtain that

$$\nu(\mathbf{k}_{\text{EP}}) = \text{sgn}(\det \begin{bmatrix} \partial_{q_x} \text{Re} \tilde{\Delta}(\mathbf{q}) & \partial_{q_y} \text{Re} \tilde{\Delta}(\mathbf{q}) \\ \partial_{q_x} \text{Im} \tilde{\Delta}(\mathbf{q}) & \partial_{q_y} \text{Im} \tilde{\Delta}(\mathbf{q}) \end{bmatrix}) = \text{sgn}(\det \begin{bmatrix} 1 & c_1^r \\ 0 & c_1^i \end{bmatrix}) = \text{sgn}(c_1^i), \quad (\text{R26})$$

where $\text{sgn}(c_1^i)$ represents the sign of the c_1^i . Note that in our statement in the ‘‘Corollary’’ section of the main text, $c_0 = \sqrt{c_x}$ and $c_1 = c_y/c_x$. Therefore, a stable EP requires a nonzero imaginary part of c_1 , which further ensures a nonzero spectral area A_{\pm} .

Our theorem states that the universal skin effect appears if and only if the spectral area is nonzero. Therefore, a corollary of the theorem can be obtained that all lattice Hamiltonians with stable exceptional points must have the universal skin effect.

 In the revised version, we have made the following changes:

- In the main text, we have modified the sentence ‘‘Here c_0, c_1 are nonzero complex numbers and $c_1 \notin \mathbb{R}$.’’ in the first paragraph of the ‘‘Corollary’’ section as ‘‘Here c_0, c_1 are nonzero complex numbers and the stable exceptional point ensures the nonzero imaginary part of c_1 .’’
- We have added the above discussion into Supplementary Note 5-C to support the statements in the main text.

‘‘(7) In Fig.3(c, d), what determine the position where the eigenstates are localized? Also, I think it is good to expand Fig.3 and include the energy spectra under periodic and open boundary condition. ’’

We found the following *inaccurate* statement in the ‘‘Corollary’’ section in our original version, ‘‘The skin effect disappears under square geometry but reappears under diamond geometry shown in Fig. 3 (c)(d).’’, which may cause some misleading to the reviewer. In fact, Fig.3(c, d) shows the time evolution of the wave-packets at different times. Therefore, the wave functions shown in Fig.3(c, d) are not the eigenstates, which has been clarified in our revision.

In the original version, Fig. 3 mainly focus on the difference of dynamical behavior between Hermitian (no skin effect) and non-Hermitian (has GDSE) cases, therefore, we think that it may be better to show the energy spectra under PBC and OBC in

FIG. R6. For the edge in red color in (a), one can choose the momentum basis as (k_x, k_y) . The corresponding spectra in E - k_x space and its projection on the complex plane are shown in (b). For the edge in red color in the diamond geometry, the momentum basis is chosen as shown in (c). The spectra in E - q_{\parallel} space and its projection on the complex plane are shown in (d).

the Supplementary Information.

We guess the concern from the reviewer is what determines the localized position of the eigenstates for the corresponding non-Hermitian Hamiltonian of Fig. 4(d) (that is the Fig. 3(d) in the original version). If so, the answer to this question has been replied in question (5.1).

 Here we directly use the method introduced in the reply of question (5.1) to analyze the results.

For each edge of the open-boundary geometry, we can transform the momentum basis from $\mathbf{k} = (k_x, k_y)^T$ to $\mathbf{q} = (q_{\parallel}, q_{\perp})^T$ by Eq.(R14). Accordingly, the Hamiltonian is transformed from $\mathcal{H}(k_x, k_y)$ to $\tilde{\mathcal{H}}(q_{\parallel}, q_{\perp})$. For the edges in red color in Fig. R6(a, c), S is chosen as $\{\{1, 0\}, \{0, 1\}\}$ and $\{\{-1, -1\}, \{0, -1\}\}$, respectively. From Fig. R6(d) we can see that, for almost all q_{\parallel} the spectral winding defined in Eq.(R15) is nonzero, consequently, the red-color edge in Fig. R6(c) has skin modes. As a comparison in Fig. R6(b), the spectral winding $w_{E_b}(k_x)$ is zero for each k_x , then the edge in red color in Fig. R6(a) has no skin modes.

 In the revised version, we have made the following changes:

- We have corrected the statement in the ‘‘Corollary’’ section
 ‘‘The skin effect disappears under square geometry but reappears under diamond geometry shown in Fig. 3 (c)(d).’’ as
 ‘‘The skin effect disappears under square geometry but reappears under diamond geometry, which is verified in the Supplementary Section VI.’’
- We have included the energy spectra under PBC and OBC in the Supplementary Note 6-A.

‘‘(8) There is a short description of the dynamical properties associated with the skin effect. Is it new? If it is the case, more detailed description is necessary. Why does the skin effect induce transverse motion of the wave packet? ’’

We are grateful for these valuable questions and helpful suggestion.

For the first question, the calculation of dynamics associated with skin effect can be traced back to Ref. [11]. However, as far as we know, there is no quantitative theory for the description of the dynamics in non-Hermitian systems with skin effect.

For the second question, we state that the skin effect causes some components of the wave packet to drift parallel to the edge in red color in Fig. R7(a), and suppresses the reflection components perpendicular to this edge. Therefore, after several bounces between the edges, the wave packet finally appears as a transverse motion shown in Fig. 4(d) in the main text.

 In what follows, we will explain the statement for the second question in two steps.

In our example, the wave packet has Gaussian form centered at $\mathbf{k}_c = (-2, -2)$ in momentum space, and evolves according to

$$|\psi_t\rangle = \mathcal{N}(t)e^{-i\mathcal{H}_{\text{OBC}}t}|\psi_0\rangle. \quad (\text{R27})$$

Therefore, the wave packet will hit the edge in red color in Fig. R7(a) and scattering off it. In this process, the momentum component of the wave packet parallel to the edge is conserved. Hence, we transform the momentum basis from (k_x, k_y) to

FIG. R7. (a) The Gaussian wave packet is centered at the diamond geometry at the initial time, and the momentum basis q_x and q_y are parallel to edge 2 and edge 1, respectively. The q_y^0 -component of the Gaussian wave packet has a plane-wave form in q_y direction and Gaussian form in q_x direction illustrated as the gray wave packet centered at $q_x = 0$ in (b)(c). The real- and imaginary-part band structures of $\mathcal{H}(q_x, q_y)$ in Eq.(R28) are shown in (b) when $\gamma = 0$, and are plotted in (c) when $\gamma = 1/4$.

(q_x, q_y) , as shown in Fig. R7(a), where $\mathbf{k} = S\mathbf{q}$ with $S = \{\{-1, -1\}, \{1, -1\}\}$. In \mathbf{q} basis, the Gaussian wave packet is centered at $\mathbf{q}_c = (0, 2)$. Accordingly, the Hamiltonian Eq.(4) in the main text can be rewritten in \mathbf{q} basis,

$$\tilde{\mathcal{H}}(q_x, q_y) = \mathcal{H}(S\mathbf{k}) = \mathbf{d}(-q_x - q_y, q_x - q_y) \cdot \boldsymbol{\sigma} - i\gamma/2(\sigma_0 - \sigma_z). \quad (\text{R28})$$

For a given q_y (here we select $q_y = q_y^0 = 3/2$), the spectral winding number $w_{E_b}(q_y^0)$ (defined in Eq.(R15)) is nonzero, which implies the existence of skin effect on the edge 1 in Fig. R7(a). Next, we will show that this skin effect results in the transverse drift of some components of the wave packet.

- **The transverse drift of the wave packet from the skin effect on edge 1**

We focus on the component that has plane-wave form in q_y direction and Gaussian form in q_x direction. Note that the actual two-dimensional Gaussian wave packet is the coherent superposition of these components with a certain weight. Here we select the component with $q_y^0 = 3/2$. In q_x direction, the component is a Gaussian wave packet centered at $q_x = 0$ as shown in Fig. R7(b)(c). In Hermitian case ($\gamma = 0$), the component will disperse with zero group velocity in q_x direction. In Fig. 3(c) of the main text, therefore, the Gaussian wave packet always slowly disperses with time in q_x direction.

However, in non-Hermitian case ($\gamma = 1/4$), the component will have a drift along the positive q_x direction. As shown in Fig. R7(c), around $q_x = 0$ each energy band has symmetric real part but asymmetric imaginary part, which corresponds to the presence of skin effect. In the range from $q_x = 0$ to 1, the energy band in blue color has a larger imaginary part and positive group velocity ($\partial \text{Re } E / \partial q_x > 0$), while in the range from $q_x = -1$ to 0, the red-color energy band has a larger imaginary part and positive group velocity. The components with larger imaginary part decay more slowly and therefore dominate the evolution of the wave packet as time goes on. Consequently, some components of the Gaussian wave packet have a transverse drift along the positive q_x direction as shown in Fig. 3(d) of the main text, which is different from the Hermitian case.

Note that this transverse drift of wave packet originates from the bulk spectral property and can occur without hitting the boundary.

Next, we will simply state that the suppression of the reflection wave components off the edge 2 ascribes to the skin effect on the edge 2 in Fig. R7(a). The general theory for describing this phenomenon will be present in our next work before long.

- **The suppression of the reflection wave components from the skin effect on edge 2**

In the process of wave packet hitting the boundary and scattering off, q_x is preserved. Therefore, we consider the wave-packet component that has plane-wave form in q_x direction and Gaussian form in q_y direction, labeled by q_x -component. The actual two-dimensional Gaussian wave packet is composed of these q_x -components with different weight. The skin

effect on edge 2 means that for a given q_x , the spectral winding number $w_{E_b}(q_x)$ is nonzero, which suppresses the reflection wave of the q_x -component.

To sum up, due to the skin effect on the edge 1 and edge 2, the Gaussian wave packet, going through several bounces between the two edges (edge 1 and the other parallel edge), finally appears as a transverse motion.

In the revised version, we have added the above discussion and Fig. R7 into Supplementary Note 6-C-1 to support the statement in the main text.

“(9) CSE in three-dimensional systems with exceptional lines should be properly described in the main text including relevant figures. Also, is it possible to observe GDSE for 3D systems with exceptional lines?”

FIG. R8. The Hamiltonian Eq. (R29) possesses four exceptional lines in 3D Brillouin zone shown in (a). The spatial distribution of eigenstates on the cube geometry ($L_x = L_y = L_z = 12$) is plotted in (b). For these two-dimensional subsystems with $k_z = 0, \pi/5, 3\pi/4$, we plot in (c) the periodic-boundary spectrum (the light blue region) and the open-boundary eigenvalues (the red dots) under the lower triangle geometry with system size $L_x = L_y = 30$. In addition, we show the spatial distribution of eigenstates on these triangle geometries.

The answer to this question is yes.

In following contents, we construct a three-dimensional model with exceptional lines, and show that it has GDSE.

Here, we construct a three-dimensional model by coupling the two-dimensional photonic crystal model (Eq.(4) in the main text) along the z direction. In this model, we can observe four exceptional lines crossing the Brillouin zone along k_z direction, meanwhile, the system exhibits GDSE. The Hamiltonian of this model reads

$$\mathcal{H}(\mathbf{k}) = \mathbf{d}(k_x, k_y) \cdot \boldsymbol{\sigma} - i\gamma/2(\sigma_0 - \sigma_z) + \cos k_z \sigma_z, \quad (\text{R29})$$

where $\boldsymbol{\sigma} = (\sigma_0, \sigma_x, \sigma_y, \sigma_z)$ is a vector of the Pauli matrices and $\mathbf{d}(k_x, k_y)$ is a vector with four components, that is,

$$\begin{aligned} \mathbf{d}(k_x, k_y) = & \{ \mu_0 - (t_2 + t_3)(\cos k_x + \cos k_y), \\ & t_1[1 - \cos k_x - \cos k_y + \cos(k_x - k_y)], \\ & t_1[\sin k_x - \sin k_y - \sin(k_x - k_y)], \\ & \mu_z + (t_2 - t_3)(\cos k_x - \cos k_y) \}. \end{aligned} \quad (\text{R30})$$

Here, we choose $\gamma = 1$, the only non-Hermitian parameter. The other parameters are the same as that in the main text, $(t_1, t_2, t_3, \mu_0, \mu_z) = (0.4, -0.1, 0.5, 1.35, -0.02)$. In this 3D model, the coupling term along z direction is $\cos k_z \sigma_z$. The Hamiltonian possesses four exceptional lines in 3D Brillouin zone, which are plotted as the blue lines in Fig. R8(a). When $k_z = \pm\pi/2$, the Hamiltonian Eq. (R29) reduces to our two-dimensional photonic crystal model with four exceptional points. Next we demonstrate that the system exhibits GDSE.

By definition, the sufficient and necessary condition for the existence of GDSE is that both of the following two points are satisfied: (i) the spectral area is nonzero; (ii) there is at least one geometry under which the skin effect disappear. The first point is satisfied due to the presence of the stable exceptional lines. The second point can be satisfied when we put the Hamiltonian on the cube geometry in Fig. R8(b), which is explained as follows.

First, the presence of the mirror symmetry, $\mathcal{H}(k_x, k_y, k_z) = \mathcal{H}(k_x, k_y, -k_z)$, forbids the skin effect along z axis of the cube geometry. Second, for each fixed k_z , $\cos k_z$ can be absorbed into μ_z , and the two-dimensional subsystem has no skin effect with square geometry. Therefore, with the cube geometry, the three-dimensional Hamiltonian does not exhibit skin effect, which is also verified by the numerical calculation. As shown in Fig. R8(b), the spatial distribution of eigenstates $W(\mathbf{x}) = \frac{1}{N} \sum_n |\psi_n(\mathbf{x})|^2$ is uniform on the cube geometry, with system size $L_x = L_y = L_z = 12$. The two points have been satisfied, therefore, the model has GDSE.

As a consequence of GDSE, the skin effect will reappear under other geometries. For example, if we cut the cube into two right-angle triangular prisms along z axis, the eigenstates will concentrate on the cross-section. Numerically, we calculate several two-dimensional subsystems with some generic k_z , and observe that skin modes will reappear under the triangular geometry, as shown in Fig. R8(c). It implies the appearance of the skin modes on the cross-sections of two triangular prisms.

 In the revised version, we made the following modifications:

- As the reviewer suggested, we have moved the example of three-dimensional exceptional-line semimetal with the corner-skin effect and related figures into the ‘‘Corollary’’ section in the revised version of the main text.
- We have added the above three-dimensional example for GDSE to Supplementary Note 6-B.

‘‘(10) It is stated that ‘‘skin effect is compatible with all spatial symmetries’’. But CSE seems to be compatible only with certain symmetries. The relevant statement should be properly modified.’’

We thank the reviewer for pointing out this confusing explanation. In the revision, we have modified the relevant statements as ‘‘the universal skin effect is compatible with all point-group symmetries’’.

 Next we explain the statement in more detail.

We want to express that the universal skin effect is compatible with all point-group symmetries in higher dimensions. The universal skin effect appears if and only if the spectral area is nonzero. And there is no point group can restrict the spectral area to be zero. Therefore, the universal skin effect can occur under any point group, that is to say, it is compatible with all point-group symmetries.

As shown in Fig. R2, if there are some point-group symmetries to restrict the current functional $J_\alpha[n] = 0$ for all α and n , the non-reciprocal skin effect (including corner-skin effect) is forbidden. However, the generalized reciprocal skin effect (for example GDSE) can appear, which is also one type of the universal skin effect in higher dimensions.

-
- [1] Zhang, K., Yang, Z. & Fang, C. Correspondence between Winding Numbers and Skin Modes in Non-Hermitian Systems. *Phys. Rev. Lett.* **125**, 126402 (2020).
- [2] Nielsen, H. B. & Ninomiya, M. Absence of neutrinos on a lattice:(I). Proof by homotopy theory. *Nuclear Physics B* **185**, 20–40 (1981).
- [3] Okuma, N., Kawabata, K., Shiozaki, K. & Sato, M. Topological Origin of Non-Hermitian Skin Effects. *Phys. Rev. Lett.* **124**, 086801 (2020).
- [4] Kawabata, K., Shiozaki, K., Ueda, M. & Sato, M. Symmetry and Topology in Non-Hermitian Physics. *Phys. Rev. X* **9**, 041015 (2019).
- [5] Yi, Y. & Yang, Z. Non-Hermitian Skin Modes Induced by On-Site Dissipations and Chiral Tunneling Effect. *Phys. Rev. Lett.* **125**, 186802 (2020).
- [6] Yao, S. & Wang, Z. Edge States and Topological Invariants of Non-Hermitian Systems. *Phys. Rev. Lett.* **121**, 086803 (2018).
- [7] Yokomizo, K. & Murakami, S. Non-Bloch Band Theory of Non-Hermitian Systems. *Phys. Rev. Lett.* **123**, 066404 (2019).
- [8] Yang, Z., Zhang, K., Fang, C. & Hu, J. Non-Hermitian Bulk-Boundary Correspondence and Auxiliary Generalized Brillouin Zone Theory. *Phys. Rev. Lett.* **125**, 226402 (2020).
- [9] Yang, Z., Schnyder, A. P., Hu, J. & Chiu, C.-K. Fermion Doubling Theorems in Two-Dimensional Non-Hermitian Systems for Fermi Points and Exceptional Points. *Phys. Rev. Lett.* **126**, 086401 (2021).
- [10] Shen, H., Zhen, B. & Fu, L. Topological Band Theory for Non-Hermitian Hamiltonians. *Phys. Rev. Lett.* **120**, 146402 (2018).
- [11] Yao, S., Song, F. & Wang, Z. Non-Hermitian Chern Bands. *Phys. Rev. Lett.* **121**, 136802 (2018).

III. RESPONSE TO REVIEWER #2

“In the manuscript “Universal non-Hermitian skin effect in two and higher dimensions”, the authors propose a universal theorem predicting the non-Hermitian skin effect in general systems solely from the spectral area covered by the complex eigenvalues in a periodic geometry. This general criterion provides a sought-after bulk-boundary correspondence and allows the distinction of two different flavours, the corner- and geometry-dependent skin effect. Following that, the authors relate the skin effect to the presence of exceptional points, potentially opening experimental verifications.

In the light of the multitude of works on higher order non-Hermitian skin effects which either relied on the presence of certain symmetries or applicability of topological invariants, this work stands out as providing a universal criterion for all of these systems. Moreover it differentiates purely geometrically from bulk induced skin effects and provides sound proofs as well as intuitive explanations for the presented statements. I find this an elegant work, and an important contribution to the field of non-Hermitian topology. I can thus recommend it for publication if the following points are clarified.”

The authors appreciate the reviewer for his/her efforts in reviewing our work, and are grateful for the positive evaluation of our manuscript, that is, “I find this an elegant work, and an important contribution to the field of non-Hermitian topology.”.

We thank the reviewer again for his/her valuable suggestions and comments on our manuscript, which are very helpful to improve the quality of our manuscript. We have carefully responded to all questions from the reviewer and revised our manuscript by following the reviewer’s suggestions.

“(1) The authors present several eigenstate distribution plots, which lack colorbars. Especially when computed with a model presented in the manuscript, specific parameters (e.g. k-grid, system size, etc.) would also be useful in all figure captions to ensure reproducibility.”

FIG. R9. Two manifestations of the universal skin effect. One is the CSE (a)-(d), the other is the GDSE (e)-(h). In (a)(b)(e)(f), the light blue regions represent the spectrum under periodic boundary, where 200×200 k-grid is used, and the red points represent the eigenvalues under different open-boundary geometries. The system size under square geometry in (c)(g) is $L_x \times L_y = 60 \times 60$, and each triangle geometry in (d)(h) has the same right-angled side length $L_x = L_y = 60$. The spatial distributions of eigenstates $W(\mathbf{x})$ are plotted in (c)(d)(g)(h) with the color bars. In the system with GDSE, the skin effect disappears under square geometry (geometry 1) in (g), and reappears under triangle geometry (geometry 2) in (h).

We thank our reviewer for these valuable and constructive suggestions.

Based on these suggestions, in the revised version, we have made the following changes:

- We have adjusted the eigenstate-distribution plots and added the corresponding colorbars for all figures in our manuscript.
- In addition, we have clearly stated the specific parameters, including k-grid and system size, in all figure captions.

An example of the revised figure is shown in Fig. R9.

Now, we believe that for each figures, we have provided a clear description to ensure reproducibility.

“(2) The authors present a corollary predicting a skin effect whenever stable exceptional points are present. This fact was already discussed in Ref.29 (Phys. Rev. Research 2, 023265), stating that the bands between two exceptional points necessarily form a point gap with non-trivial 1D winding number, therefore showing a skin effect under open boundaries. I think the authors should highlight this work, even though their approach does not rely on a topological invariant.”

We thank our reviewer for pointing out this point. We have revisited the paper and found that this work indeed has a close connection with our corollary, and decide to highlight this work.

In the revised version, we have highlighted this work in the “Corollary” section. To be more precise, we have added the following statement after the first sentence of this section,

“This connection has also been discussed in Ref. [29], where the bands around the stable exceptional point form a point gap with nonzero spectral winding number, consequently, exhibiting the skin effect under an open-boundary geometry.”

“(3.1) A feature of the GDSE seems to be that the area of the OBC spectrum is the same as in the PBC case. Can the authors comment on whether this is an artefact or feature, i.e. can one proof this relationship?”

FIG. R10. (a) The triangle geometry. (b) The red points represent the eigenenergies of the Hamiltonian \mathcal{H} (Eq.(3) in the main text) under triangle geometry with different system size; the light blue area is the periodic-boundary spectrum $\mathcal{H}(BZ)$.

We believe that it is a universal feature of the GDSE based on the numerical results. However, unfortunately, we can not strictly prove this relationship.

Now we show some numerical results to demonstrate this point.

As shown in Fig. R10, one can observe that as the system size increases, the area covered by the open-boundary spectrum on the complex plane becomes gradually larger. It can be expected that under the thermodynamic limit, the region of the open boundary spectrum tends to be the same as the region covered by the Bloch Hamiltonian spectrum (the light blue region in Fig. R10)

In the revised version, the following changes has been made:

- We have adjusted the statement

“Besides the distribution of the eigenstates, another feature of the GDSE is that area of the open-boundary spectrum seems to be the same as A_i . However, the corresponding density of states is dependent by the choice of geometry as shown in Fig. 2 (e)(f).”

in the last paragraph of the section “The corner-skin and the geometry-dependent-skin effect” as

“In one dimension, an open chain does not exhibit skin effect when its spectrum coincides with the corresponding periodic-boundary spectrum on the complex plane. Unlike in one dimension, even if the region covered by the energy spectrum under some open-boundary geometry (such as the triangle geometry in Fig. 3(h)) seems to be the same as the region covered by the periodic-boundary spectrum, the system can still show a skin effect due to the different density of states on the complex plane. It is also a unique feature in two- and higher-dimensional skin effects.”

- The related numerical results have been added into the Supplementary Note 4-B-3.

“(3.2) Also I would suggest to complement Fig. 2 e,f with a density of states plot to supplement the discussion in the main text.”

FIG. R11. The energy spectra of the Hamiltonian (Eq.(3) in the main text) under periodic boundary condition, square and triangle geometry are shown in (a-c), respectively, and the corresponding density of states on the complex plane is shown in (d-f). In (a)(d), the 60×60 k -grid is used. In (b)(e) and (c)(f), the system size under square geometry (geometry 1 in Fig. R9) and triangle geometry (geometry 2 in Fig. R9) is taken as $L_x = L_y = 60$ and $L_x = L_y = 85$, respectively.

We thank our reviewer again for this constructive suggestion.

In the revise version, the following changes have been made:

- We have added Fig. R11 into the Supplementary Note 4-B-3.
- An explanation of Fig. R11, which is shown below, has also been provided in the Supplementary Note 4-B-3.

“We complement the density of states plot to support the statement in the main text. In Fig. R11, the open-boundary spectra under periodic boundary, square open-boundary geometry and triangle open-boundary geometry are shown in (a)(b)(c), respectively. The corresponding density of states on the complex energy plane are plotted in (d-f), where the z-axis $D(E)$ indicates the probability density of eigenvalues lying in the unit energy interval at E on the complex plane. The spectra under periodic boundary condition and square geometry have the same density of states on the complex energy plane as shown in (d) and (e), where the spectral distribution at the corner is denser than the center. It is not the case on the triangle geometry. Even though the same area is covered by their spectra, the eigenvalues under the triangle geometry are more densely distributed at the center of the spectra shown in (f).”

“(4) The authors provide a proof that the GDSE is absent in certain geometries. Is there a way to find the geometries not showing a skin effect without trying brute-force? This could be very useful in practical settings.”

We are grateful to our reviewer for raising this important question. Our reply to this question is that for the symmetry-protected-GDSE (see the following explanation), the answer is yes, and the required geometry not showing a skin effect depends

exactly on the bulk symmetry. However, for the GDSE with no relation to the symmetries, we have not found an efficient method to find the geometries not showing a skin effect.

In what follows, we will show how to find the corresponding geometry when the symmetries of the bulk Hamiltonian are given. Here we will only use an example with mirror symmetry to demonstrate the procedure, a more complete discussion will appear in our forthcoming paper.

Mirror Symmetry (\mathcal{M})

If the bulk Hamiltonian $\mathcal{H}(\mathbf{k})$ has one mirror symmetry, e.g., the mirror- x symmetry $\mathcal{M}_x \mathcal{H}(k_x, k_y) \mathcal{M}_x^{-1} = \mathcal{H}(-k_x, k_y)$, the (vertical) boundaries parallel to the mirror line does not exhibit the skin effect due to the spectral winding number being zero,

$$\begin{aligned} w_{E_b}(k_y) &= \frac{1}{2\pi i} \int_{-\pi}^{\pi} dk_x \partial_{k_x} \log \det[\mathcal{H}(k_x, k_y) - E_b] = \frac{1}{2\pi i} \int_{-\pi}^{\pi} dk_x \partial_{-k_x} \log \det[\mathcal{H}(-k_x, k_y) - E_b] \\ &= \frac{1}{2\pi i} \int_{-\pi}^{\pi} dk_x \partial_{-k_x} \log \det[\mathcal{M}_x \mathcal{H}(k_x, k_y) \mathcal{M}_x^{-1} - E_b] = -w_{E_b}(k_y) = 0, \end{aligned} \quad (\text{R1})$$

regardless of the reference energy E_b . If the bulk Hamiltonian has another mirror symmetry, for example, the mirror- y symmetry $\mathcal{M}_y \mathcal{H}(k_x, k_y) \mathcal{M}_y^{-1} = \mathcal{H}(k_x, -k_y)$, the (horizontal) boundaries parallel to the mirror- y line does not show the skin effect for the same reason. Therefore, we conclude that if the bulk Hamiltonian has two or more mirror symmetries, the skin effect does not appear on the open-boundary geometry with each boundary parallel to one of these mirror lines (one example is Fig. 2(g) in the original main text).

In the revised version, the above discussion has been added in the Supplementary Note 4-B-1 to explain the vanishing of skin effect in Fig. 3 (g) of the main text.

“(5) In order to highlight the influence of the skin effect on dynamical properties, the authors simulate the evolution of a wavepacket in a geometry with exceptional points. Since driven/dissipative systems are usually governed by the time-dependent Lindblad equations, could the authors comment on why the Hermitian time evolution still holds in their setting?”

We thank the reviewer for this insightful and important question. First, we expect that the anomalous dynamical behavior in the wave-packet simulation [Fig. 4(d) of the main text] can be observed in a realistic photonic crystal with gain and/or loss. Second, in dissipative open quantum system, it can be shown that the evolution of the mean value of the particle number for a given prepared initial state evolves under an effective non-Hermitian Hamiltonian, which has the same form as the Hermitian case (a closed quantum system).

The above two points will be explained in detail as follows.

In classical wave system, the evolution of a system state is governed by wave equation formally analogous to the Schrödinger equation [1]. In the cases with gain and/or loss, like the photonic crystal model in our manuscript, the time evolution of a localized excitation (wave packet) is nonunitary, which can be captured by an effective non-Hermitian matrix [2]. Therefore, we believe that the phenomena in the simulation of wave-packet dynamics, e.g., the anomalous dynamical behavior shown in Fig. 4 of the main text, can be observed in a realistic classical wave system.

In the (driven or dissipative) open quantum system, a system state is described by the density matrix $\hat{\rho}$, whose time evolution is governed by the Lindblad master equation [3],

$$\frac{d\hat{\rho}}{dt} = -i[\hat{H}, \hat{\rho}] + \sum_x (2\hat{L}_x \hat{\rho} \hat{L}_x^\dagger - \{\hat{L}_x^\dagger \hat{L}_x, \hat{\rho}\}). \quad (\text{R2})$$

Here $\hat{H} = \sum_{xy} \mathcal{H}_{xy} \hat{c}_x^\dagger \hat{c}_y$ is the system Hamiltonian, and $\hat{L}_x = \sum_y \mathcal{D}_{xy} \hat{c}_y$ is the Lindblad dissipators describing quantum jumps due to coupling to the environment. In this setting, the dynamics of the density matrix $\hat{\rho}(t)$ can be captured by the single-particle correlation $C_{xy}(t) = \text{tr}[\hat{\rho}(t) \hat{c}_x^\dagger \hat{c}_y]$. After some tedious calculations, one obtains that the time evolution of the correlation follows [4, 5]

$$C(t) = G(t)C(0)G^\dagger(t); \quad G(t) = e^{i\mathcal{H}_{\text{eff}}t}, \quad (\text{R3})$$

where $\mathcal{H}_{\text{eff}} = (\mathcal{H} - i\mathcal{D}^\dagger\mathcal{D})^t$ is the effective non-Hermitian matrix, the superscript “t” representing the transpose of the matrix. In the closed (Hermitian) quantum system, the system has no coupling with the external environment, and the density matrix evolves according to $d\hat{\rho}(t)/dt = -i[\hat{H}, \hat{\rho}]$. As a result, the dynamics of correlation satisfies

$$C(t) = G(t)C(0)G^\dagger(t); \quad G(t) = e^{i\mathcal{H}^t t}, \quad (\text{R4})$$

where \mathcal{H}^t is transpose of the Hamiltonian matrix \mathcal{H} . Comparing Eq.(R3) with Eq.(R4), we conclude that the time evolution of the correlation $C(t)$ in dissipative open quantum system has the same form as the Hermitian case.

In the revised version, we have added the above discussion in the Supplementary Note 6-C-2.

“Small remark: Please capitalize “Hamiltonian” throughout the manuscript. ”

Thanks the reviewer for pointing out this typographical error, which has been corrected in the revised version.

-
- [1] Özdemir, Ş. K., Rotter, S., Nori, F. & Yang, L. Parity–time symmetry and exceptional points in photonics. *Nature Materials* **18**, 783–798 (2019).
 - [2] Ashida, Y., Gong, Z. & Ueda, M. Non-hermitian physics. *Advances in Physics* **69**, 249–435 (2020).
 - [3] Dalibard, J., Castin, Y. & Mølmer, K. Wave-function approach to dissipative processes in quantum optics. *Phys. Rev. Lett.* **68**, 580–583 (1992).
 - [4] Song, F., Yao, S. & Wang, Z. Non-Hermitian Skin Effect and Chiral Damping in Open Quantum Systems. *Phys. Rev. Lett.* **123**, 170401 (2019).
 - [5] Mao, L., Deng, T. & Zhang, P. Boundary condition independence of non-Hermitian Hamiltonian dynamics. *Phys. Rev. B* **104**, 125435 (2021).

IV. RESPONSE TO REVIEWER #3

“Review on “Universal non-Hermitian skin effect in two and higher dimensions” by Kai Zhang, Zhesen Yang, and Chen Fang.

First, I summarize what the authors study in this paper and new finding. In non-Hermitian systems, one of the important issues is the non-Hermitian skin effect (NHSE), where the spectrum strongly depends on the boundary conditions. For one-dimensional systems, it was shown in Refs. [17,18] that the nonzero winding number of complex spectra under the periodic boundary condition is a sufficient condition for the existence of the NHSE under the open boundary condition. In more general spatial dimensions, many examples of non-Hermitian Hamiltonians show a sort of NHSEs, and a systematic understanding of the NHSE in higher dimensions is required. In this paper, the authors propose a necessary and sufficient condition for the NHSE, described below. The central claim of this paper is that the NHSE occurs under any open boundary condition if and only if the complex spectrum has a finite area under the periodic boundary condition. Such a general condition for the NHSE in high dimensions has not yet been proposed except in this paper, and I think it is an impactful result.

While the results of this paper are significant and timely, the quality of the paper may not be up to the level of publication in the following points. ”

We thank the reviewer for the careful reading of our manuscript and the positive comment, namely “Such a general condition for the NHSE in high dimensions has not yet been proposed except in this paper, and I think it is an impactful result. ”.

Regarding the concerns of the reviewer, we have made a serious response to every point in the following contents. We are now confident that we have cleared the reviewer’s doubts. In addition, we thank the reviewer for his/her insightful questions and suggestions, which are of great help in improving the rigor and quality of our manuscript.

“(1) First of all, the main body of the paper says that it has proved a certain claim, but if we read the proof details of the supplemental material, it turns out that it depends partly on the following conjecture: When the winding number along a direction in the BZ is non-zero, the boundary perpendicular to that direction results in the NHSE. As the authors also pointed out in the Supplementary material, if all spatial directions are in open boundary conditions, and this is the situation of interest in the paper, this is still a conjecture. However, the main body of the paper does not explicitly state that part of the proof depends on the conjecture, which may cause misunderstanding among readers. ”

FIG. R12. The sketch of the proof of the theorem.

We are grateful to the reviewer for pointing out this key issue. Indeed, the proof of our “theorem” does depend on a conjecture, and in the previous version, we only mentioned this in the appendix.

During the preparation of the reply, we tried again to prove the above conjecture but failed. We now believe that the above conjecture is deeply related to the two-dimensional generalized Brillouin zone theory, the formalism of which is still an open question.

Based on the above reasons, the modifications focus on (i) emphasizing that the proof of our theorem relies on a conjecture and (ii) providing some numerical verifications of the conjecture and explaining why this conjecture makes sense.

 In summary, based on the reviewer's suggestion, in the revised version, we have made the following modifications:

- In the main text, we have added Fig. R12 and a corresponding explanation to explicitly show that the proof of the theorem depends on a conjecture.

We hope that the above figure can help the readers understand the outline and the limitation of our proof more clearly.

- In the Supplementary Information, we divided Supplementary Note 1-B (which corresponds to the proof of our theorem) into three parts, which correspond to the steps I, II, and III in Fig. R12, respectively.

Specifically, Step I and II have been strictly proved in the first two parts. In the third part, we argued that the proof of the step III is based on the conjecture. We further explained why the conjecture makes sense, and provided some numerical calculations to support our conjecture.

Through the above modifications, we hope to make it clear that the strict proof of our theorem depends on a conjecture. At the same time, we also expect that our revised version can inspire one or more readers to prove our conjecture more strictly.

 In what follows, we summarized the changes in the revision in detail, which includes the improvement of the strictly proved part and the conjecture part.

- **The rigorous proof of step I and II in Fig. R12: spectral area, spectral winding and skin effect on stripe geometry**

In Sec. I B of the original Supplementary Information, we have proved that if the spectral area is zero, the spectral winding number along each direction in the Brillouin zone (BZ) is zero, and vice versa; if the spectral area is nonzero, there is at least one direction along which the spectral winding number is nonzero.

Now we transform the momentum basis from $\mathbf{k} = (k_x, k_y)^T$ to $\mathbf{q} = (q_{\parallel}, q_y)^T$ by

$$\mathbf{q} = S \mathbf{k}; \quad \det S = 1, \quad (\text{R1})$$

where S is a $2 * 2$ integer matrix due to the lattice momentum in the BZ is discrete. For any direction in the BZ, one can always choose the appropriate \mathbf{q} basis such that under this \mathbf{q} basis, each straight line in this direction can be obtained by fixing q_{\parallel} and running q_y from 0 to 2π . Accordingly, the Hamiltonian $\mathcal{H}(k_x, k_y)$ can be transformed into $\tilde{\mathcal{H}}(q_{\parallel}, q_y)$. For any fixed q_{\parallel} , the spectral winding can be expressed as

$$w_{E_b}(q_{\parallel}) = \frac{1}{2\pi i} \int_{-\pi}^{\pi} dq_y \partial_{q_y} \log \det[\tilde{\mathcal{H}}(q_{\parallel}, q_y) - E_b], \quad (\text{R2})$$

where E_b is the reference energy.

According to what we have proved, if the spectral area is zero, under any transformation S , the spectral winding $w_{E_b}(q_{\parallel}) = 0$ for any q_{\parallel} and E_b . It means that the Hamiltonian on any stripe geometry does not exhibit skin effect if spectral area is zero. Here the stripe geometry refers to the geometry with open boundary in only one direction and periodic boundary in other direction (preserves the momentum q_{\parallel}). If the spectral area is nonzero, there is at least one stripe geometry on which the Hamiltonian shows skin effect.

- **Step III in Fig. R12: The conjecture**

Conjecture: For a given edge, if it does not exhibit skin effect under any types of stripe geometry, then it does not show skin effect under any fully open boundary geometry, vice versa — if it does not show skin effect under any fully open boundary geometry, then it does not exhibit skin effect under any types of stripe geometry.

Note that the above statement is equivalent to the conjecture in step III of Fig. R12, that is, skin effect on stripe geometry implies skin effect on fully open-boundary geometry (i.e., the universal skin effect).

Next we explain why the conjecture makes sense, and show some numerical results to support our conjecture.

- First, we believe that whether a given edge exhibits a skin effect or not depends only on the bulk spectral topology in the direction perpendicular to this edge, having nothing to do with other boundary conditions, which accords with the spirit of bulk-boundary correspondence.
- Second, for a given edge, no skin effect means that the wave packet obeys the conventional law of reflection on this edge (as shown in Fig.4(c) of the main text). Otherwise, the skin effect leads to the anomalous dynamical behavior on this edge (as shown in Fig.4(d)). This local physical consequence caused by the skin effect can not be affected by any change to other edges in the thermodynamic limit. Therefore, we conjecture that if a given edge under stripe geometry does not exhibit the skin effect, it still does not show the skin effect under any open boundary geometry, and vice versa.

FIG. R13. Some numerical examples to support the conjecture. The lower edge has always no skin effect, regardless of the shape of the fully open-boundary geometry.

- Third, we show the numerical results in Fig. R13 to support the conjecture. The bulk Hamiltonian of this model reads

$$\mathcal{H}(\mathbf{k}) = 2 \cos k_x + i \cos k_y. \quad (\text{R3})$$

The mirror- y symmetry of the bulk Hamiltonian makes the zero spectral winding for any a k_x -subsystem. As a result, the Hamiltonian on the stripe geometry with only open boundary in y direction does not show the skin effect. Even if opening the boundary in x direction and deforming the geometry into some generic open-boundary geometries, there is still no skin effect on the y -directional open boundary, as shown in Fig. R13.

“(2) Related to the comment above, since this paper claims to have proved a certain theorem, I think the outline of the proof should be written in the main text.”

We thank our reviewer again for this constructive suggestion that can improve the quality of our manuscript.

As the reviewer suggested, we have made the following changes in revised version of the main text:

1. We have added the Fig. R12 into the main text.
2. The following statements have been added after the third paragraph in the “Theorem” section,

“A brief outline of the proof is illustrated in Fig. 2. The theorem is obtained in three steps: step I establishes the equivalence relation between spectral area and spectral winding number of straight lines in the BZ; step II connects these nonzero spectral winding numbers with skin effect on the stripe geometry — the geometry with open boundary in only one direction and periodic boundary in other directions; step III illustrates that skin effect on stripe geometry implies skin

effect on fully open-boundary geometry (i.e., the universal skin effect), which is our conjecture. The justification of this conjecture is discussed in the Supplementary Note 1. ”.

“(3) The theorem claims that a finite area in the complex spectrum causes the NHSE in *any* open boundary condition. However, GDSE says that there is at least one geometry where the NHSE disappears, which contradicts the claim of the theorem. Therefore, the statement of the theorem should be refined so that it is consistent with GDSE. ”

We are grateful to our reviewer for this helpful suggestion. In the previous version, our theorem is claimed as follows:

“Now we are ready to state the theorem of universal skin effect: in the thermodynamic limit, the skin effect is present in a Hamiltonian having open boundary of arbitrary geometry, if the spectral area is nonzero ($A_i \neq 0$); vice versa, the skin effect is absent for all possible geometries, if the spectral area is zero ($A_i = 0$).”

In our original intention, the “arbitrary geometry” mentioned in the theorem refers to “a given open boundary with arbitrary (which refers to *non-special and sufficient irregular*) geometry.” However, as pointed by this question from the reviewer, this wording “arbitrary geometry” may cause confusion and be mistaken for “any open boundary geometry”.

In the revision, we have refined the statement of the theorem as “the skin effect is present in a Hamiltonian having open boundary of *generic* geometry, if the spectral area is nonzero.”. Now we believe that this statement is more precise than the previous one, and is consistent to existence of GDSE.

In addition, we adjusted the figure caption for Fig. 1 (b)(d) as “(b)(d) shows that if the spectral area of $\mathcal{H}(\mathbf{k})$ is nonzero, the skin effect will appear on some generic open-boundary geometries. ”. Also, other related statements in the manuscript have been corrected.

“(4) For the CSE, there are only examples and no definition of CSE. Sec. IIIB of the Supplemental material shows that it is equivalent to the current functional being non-zero, but how do we define it as a phenomenon?”

(5) Also, there is no definition for GDSE, only examples. ”

FIG. R14. The universal skin effect can be further classified into two types by the current functional, that is, non-reciprocal skin effect ($\exists \alpha, n; J_\alpha[n] \neq 0$) and generalized reciprocal skin effect ($\forall \alpha, n; J_\alpha[n] = 0$). CSE and GDSE are representative one in these two types skin effect, respectively.

We thank the reviewer for the insightful question and valuable comments, which are very helpful to increase the rigor and improve the quality of our manuscript. Next, we will answer these two questions (4) and (5) together due to their strong correlation.

As the concern from our reviewer, the current functional being nonzero is indeed not equivalent to the *phenomenon* that almost all system eigenstates concentrate at corners of the open-boundary geometry. However, we define both of these two as the corner-skin effect, which is the lack of rigor in our previous version. Thanks for the two questions from the reviewer, we now have an overall revision that clearly defines the CSE and GDSE.

Before defining CSE and GDSE, we need to explain the current functional, including the “nonzero current functional” and “zero current functional”.

In our revised version, the current functional is rewritten as

$$J_{\alpha}[n] = \sum_{\mu} \oint_{\text{BZ}} dk^d n(E_{\mu}, E_{\mu}^*) \nabla_{\alpha} E_{\mu}(\mathbf{k}), \quad (\text{R4})$$

under the periodic-boundary condition, where μ represents the energy band index and ∇_{α} is the directional derivative along certain direction $\alpha = \sum_{i=1}^d \alpha_i \hat{e}_{k_i}$ in d -dimensional momentum space (Here \hat{e}_{k_i} represents the i -th basis in momentum space). The current functional $J_{\alpha}[n]$ is the function of the distribution function $n(E, E^*)$, and different input $n(E, E^*)$ gives different output $J_{\alpha}[n]$ with fixed α .

The nonzero current functional (labeled by $J \neq 0$) is defined as: $\exists \alpha, n, J_{\alpha}[n] \neq 0$; and as a complementary set, the zero current functional (labeled by $J = 0$) is defined as: $\forall \alpha, n, J_{\alpha}[n] = 0$. By definition, the nonzero current functional and zero current functional are complete and mutually exclusive mathematically. Therefore, we can classify the universal skin effect (nonzero spectral area) into two types according to the current functional, i.e., the non-reciprocal skin effect ($J \neq 0$) and generalized reciprocal skin effect ($J = 0$), as illustrated in Fig. R14. Note that this classification of skin effect according to the current functional is different from the classification of intrinsic point-gap topology for symmetry class [1, 2]. Please see Supplementary Note 3 for more details.

Now we clarify the CSE and GDSE.

We define the CSE as a type of the non-reciprocal skin effect ($J \neq 0$) that exhibits the particular *phenomenon* that almost all eigenstates are localized at corners of the open-boundary geometry. Similar to the definition of CSE, the GDSE is one type of generalized reciprocal skin effect ($J = 0$) showing the unique *phenomenon* that there is at least one fully open boundary geometry under which the skin effect does not appear. The clear relation between current functional and skin effect in two and higher dimensions is shown in Fig. R14.

Therefore, the CSE inherits the features of the non-reciprocal skin effect, including nonzero current functional and incompatibility with all point groups except for $\{C_m, C_2, C_3, C_4, C_6, C_{2v}, C_{3v}, C_{4v}, C_{6v}\}$, which has been rigorously proved in the Supplementary Note 3-C-1. Likewise, the GDSE inherits the features of the generalized reciprocal skin effect, including the zero current functional and compatibility with all point-group symmetries.

According to the reviewer's concerns and suggestions, in the revised version, we have made the following changes:

- We have added Fig. R14 into Fig. 3 of the main text to illuminate the relationship between the current functional and two- and higher-dimensional skin effect.
- We have given more explanation about the current functional, CSE and GDSE in section "The corner-skin effect and geometry-dependent-skin effect" in the main text.
- We have explained the current functional and skin effect in detail in the Supplementary Note 3, and described the characteristics of CSE and GDSE in the Supplementary Note 4.

Now we believe that the definition of CSE and GDSE is clear in our revised version. In addition, we show the advantages of the current functional, including classifying the skin effect and investigating the restriction of point-group symmetries on the higher-dimensional skin effect.

In the following contents, we explain why we name these two types of skin effect as non-reciprocal skin effect and generalized reciprocal skin effect.

- The non-reciprocal skin effect ($J \neq 0$).

We call the universal skin effect with nonzero current functional as non-reciprocal skin effect (NRSE), because this type of skin effect is incompatible with inversion or anomalous time-reversal symmetry [1, 3]. Here the anomalous time-reversal symmetry refers to $\mathcal{T}^{-1} \mathcal{H}^t(\mathbf{k}) \mathcal{T} = \mathcal{U}_T^{\dagger} \mathcal{H}^t(\mathbf{k}) \mathcal{U}_T = \mathcal{H}(-\mathbf{k})$. Equivalently, the inversion and anomalous time-reversal symmetry always requires zero current functional (the rigorous proof is provided in the Supplementary Note 3-C). Therefore, the NRSE is similar to the 1D skin effect that can be forbidden by the inversion or anomalous time-reversal symmetry.

- The generalized reciprocal skin effect ($J = 0$).

We name the type of universal skin effect having zero current functional as generalized reciprocal skin effect (GRSE) based on two reasons. The first reason is that this type of skin effect is compatible with the inversion or anomalous time-reversal symmetry. Note that an analog of GRSE in one dimension is the Z2 skin effect [2], which can appear when the bulk Hamiltonian respects the spinful anomalous time-reversal symmetry ($\mathcal{T}^2 = -1$). The second reason is stated as follows.

A reciprocal system requires the Hamiltonian to satisfy $\mathcal{H}^t = \mathcal{H}$ ($\mathcal{H}^t(\mathbf{k}) = \mathcal{H}(-\mathbf{k})$ in momentum space) [4], which must result in zero current functional. But generally, the zero current functional means more. For example, a Hamiltonian satisfying $\mathcal{H}^t(\mathbf{k}) = \mathcal{H}(\mathbf{k} + \mathbf{k}_\theta)$ ($\mathbf{k}_\theta \neq 0$ and $\mathbf{k}_\theta \neq -2\mathbf{k}$) also gives rise to zero current functional. A simple model Hamiltonian reads $\mathcal{H}(\mathbf{k}) = 2 \cos k_x + i \sin k_y$, which satisfies $\mathcal{H}^t(k_x, k_y) = \mathcal{H}(-k_x, \pi - k_y)$ and has zero current functional. It can be seen that the zero current functional includes but is not limited to the reciprocal skin effect [4]. Therefore, we term the type of universal skin effect with zero current functional as the “generalized” reciprocal skin effect.

“(6.1) Regarding (ii) of ”Universality,” while the universal NHSE is claimed to be compatible with spatial and time-reversal symmetries, the only symmetry discussed in Supplemental Material is unitary point group symmetry and there is no discussion of the case where time-reversal symmetry is included. Also, what kind of time-reversal symmetry should we consider? For example, in [11], a ”time-reversal symmetry” as the transpose is introduced. ”

We thank the reviewer for pointing out the vague definition of time-reversal symmetry, which may cause some misunderstandings. In the main text, the word “time-reversal symmetry” refers to the collection of the two different time-reversal symmetries, namely, the “conventional complex-conjugate-type” one \mathcal{T} and the “anomalous transpose-type” one $\tilde{\mathcal{T}}$. Here we claim that the universal NHSE is compatible with both \mathcal{T} and $\tilde{\mathcal{T}}$.

For the reviewer’s question, “what kind of time-reversal symmetry should we consider?”, the answer is the anomalous time-reversal symmetry $\tilde{\mathcal{T}}$. This is because in one-dimension, the NHSE is compatible with \mathcal{T} but incompatible with spinless anomalous time-reversal symmetry ($\tilde{\mathcal{T}}^2 = +1$) [1, 3]. As a result, only $\tilde{\mathcal{T}}$ should be considered.

According to the concerns from the reviewer, we have made the following modifications in the revised version,

- In the main text, in order to clarify the meaning of the time reversal symmetry, we change the sentence “the skin effect is, unlike in one dimension, compatible with all point-group symmetries and time-reversal symmetry” in the forth paragraph in the “Introduction” section as “the skin effect is, unlike in one dimension, compatible with all point-group symmetries and time-reversal symmetry, including complex-conjugate-type and transpose-type time-reversal symmetry in Ref. [1]”.
- In addition, we have added the discussion about the time-reversal symmetry after the case with the unitary point-group symmetry in the Supplementary Note 3-C-2.

Next, we explain why the spinless time reversal symmetry is compatible with the higher dimensional skin effect.

The spinless anomalous time-reversal symmetry (aTRS) restricts the Hamiltonian to satisfies

$$\mathcal{T}^{-1}\mathcal{H}^t(\mathbf{k})\mathcal{T} = \mathcal{U}_T^\dagger\mathcal{H}^t(\mathbf{k})\mathcal{U}_T = \mathcal{H}(-\mathbf{k}); \quad \mathcal{T}^2 = 1. \quad (\text{R5})$$

In the Ref. [3], the authors point out that this type of time-reversal symmetry restricts the periodic-boundary spectrum to be an arc-form on the complex plane, which means the absence of skin effect. Therefore, the 1D skin effect is incompatible with the symmetry in Eq.(R5).

However, it is not the case in higher dimensions. First, the spinless aTRS cannot require the spectral area to be zero, therefore, allows the existence of the universal skin effect. That is to say, the universal skin effect is compatible with the spinless aTRS. Second, the spinless aTRS requires zero current functional ($J = 0$), which is proved in Supplementary Note 3-C-2. It means that the non-reciprocal skin effect (including CSE) in Fig. R14 is incompatible with the spinless aTRS, instead, the generalized reciprocal skin effect (including GDSE) is compatible with this symmetry.

To sum up, the universal higher-dimensional skin effect is compatible with this type of time-reversal symmetry in Eq.(R5), while the one-dimensional skin effect is not.

“(6.2) Also, are there any restrictions on other kinds of internal symmetries (e.g., pseudo Hermiticity)?”

This is a good question that should be clarified in detail.

According to our theorem, the universal skin effect disappears if and only if the spectral area is zero. Therefore, in order to discuss the role of symmetries on the universal skin effect, we need only to consider under these symmetries whether the spectral area is zero or not. However, there are no such internal (or point-group) symmetries to restrict the spectral area to be zero.

Back to the question of the reviewer, once the system has pseudo Hermiticity $\eta\mathcal{H}^\dagger(\mathbf{k})\eta^{-1} = \mathcal{H}(\mathbf{k})$ [1], the eigenvalues of the Hamiltonian must be real or come in complex-conjugate pairs. It is possible for the system to have entire real Bloch spectrum, under which the spectral area is zero and the universal skin effect disappears. A particular example is $\eta = 1$, which reduces to the Hermitian case. On the other hand, once not all the Bloch spectrum are real but come in complex-conjugate pairs to cover a finite area on the complex plane, the spectral area becomes nonzero, and consequently, the universal skin effect is present.

The above discussion have been added into the Supplementary Note 3-C-3 in the revised version.

“(7) Also related to ”Universality” (ii), the discussion in Supplemental Material III-B-2 seems to be limited to three spatial dimensions. How do we understand that two-fold rotation and inversion symmetry are the same symmetry in two dimensions (except for the factor system), but the only two-fold rotation is compatible with universal NHSE?”

In the original version, we discussed the restriction of three-dimensional point groups on the current functional. The same procedure can be done in the two spatial dimensions. The universal skin effect can appear if and only if the spectral area is nonzero. Consequently, the universal NHSE is compatible with all point groups in higher dimensions, including two-fold rotation in 3D and inversion symmetry in 2D, because these point groups cannot restrict the spectral area to be zero. However, the non-reciprocal skin effect, as one type of universal NHSE shown in Fig. R14, is compatible with two-fold rotation in 3D but incompatible with the inversion in 2D.

Next, we explain why the non-reciprocal skin effect is compatible with two-fold rotation in 3D but incompatible with the inversion in 2D.

We state that in three dimensions the two-fold rotation is compatible with non-reciprocal skin effect, because the current functional $J_\alpha[n]$ can be nonzero along α (a vector in 3D BZ) parallel to the rotation axis. Note that $J_\alpha[n]$ is zero when α is perpendicular to the rotation axis. For example, consider a system with two-fold rotation symmetry along the z -axis. This rotation symmetry requires the current functional to satisfy $J_x[n] = J_y[n] = 0$ but $J_z[n] \neq 0$ (see detailed derivations in Supplementary Note 3-C-1), where

$$J_{i=x,y,z}[n] = \sum_\mu \oint_{\text{BZ}} dk^d n(E_\mu, E_\mu^*) \partial_{k_i} E_\mu(\mathbf{k}); \quad J_\alpha[n] = \sum_i \alpha_i J_i[n], \quad (\text{R6})$$

where the $\{\alpha_x, \alpha_y, \alpha_z\}$ is the representation of α under the three-dimensional momentum space. Therefore, the current functional $J_\alpha[n] = \alpha_x J_x[n] + \alpha_y J_y[n] + \alpha_z J_z[n] = \alpha_z J_z[n] \neq 0$, that is “nonzero current functional” ($J \neq 0$). Based on this, we expect that the non-reciprocal skin modes concentrate on the surface perpendicular to the rotation axis. Therefore, the non-reciprocal skin effect (one type of universal NHSE) can appear on the three-dimensional system having two-fold rotation symmetry.

The two-fold rotation symmetry reduces to inversion in two dimensions, which requires zero current functional. According to the definition in Fig. R14, the non-reciprocal skin effect cannot appear in the two-dimensional system having inversion symmetry (instead, the generalized reciprocal skin effect can exist).

In conclusion, the non-reciprocal skin effect can appear in the three-dimensional system with C_2 symmetry but cannot appear in the two-dimensional system having inversion symmetry.

In the revised version, we have added the above discussion into the Supplementary Note 3-C-1.

“I also have several minor comments listed below.

(8) I think it is worth stating that the open boundary condition in a two-dimensional system in the claim of the theorem is the full open boundary condition for all directions. ”

As suggested by our reviewer, we have added a sentence after stating the theorem in the “Theorem” section, that is, “The open boundary in the theorem refers to the fully open boundary condition for all directions.”.

“(9) Regarding NHSE of interest in the paper, I would suggest commenting somewhere that the number of skin modes follows a volume law, for example, to clarify the difference from the higher-order skin effects. ”

Thanks to the reviewer for this helpful suggestion.

In the revised version, we have added the following comment in the last paragraph of the “Theorem” section, “Additionally, the number of skin modes in the universal skin effect follows a volume law, which differentiates from the higher-order-skin effect. ”.

“(10) Regarding Fig. 2 d, the upper and lower sites of the geometry are considered to be identical, so I think the ”triangle geometry” may be misleading. ”

Fig. 3 d (namely Fig. 2 d in the original version) shows two unattached triangle open-boundary geometries with the same system size $L_x * L_y/2$, obtained by cutting the square open-boundary geometry in Fig. 3 c along the diagonal. In the revision, we also have used two colorbars in Fig. 3 d to indicate the spatial distribution of eigenstates under these two triangle open-boundary geometries.

“(11) On the left part of page 4, is the estimation for A_{\pm} correct? If the imaginary part of c_1 is infinitesimal, then the area A_{\pm} is also infinitesimal. ”

We thank the reviewer for raising this valuable question.

 Yes, the estimation for A_{\pm} is correct. In short, a *stable exceptional point* requires that the imaginary part of c_1 cannot vanish, which further necessitates a nonzero spectral area. The reason is as follows.

Here the “stable” exceptional point refers to the exceptional point whose topological charge (discriminant number) is ± 1 [5]. The Bloch Hamiltonian of a two-band model can be written as

$$\mathcal{H}(\mathbf{k}) = \sum_{i=x,y,z} h_i(\mathbf{k})\sigma_i, \quad (\text{R7})$$

where $h_i(\mathbf{k}) = h_i^r(\mathbf{k}) + ih_i^i(\mathbf{k})$. Here we have omitted the $h_0(\mathbf{k})\sigma_0$ term, which is irrelevant to the discussion of exceptional points. The eigenvalues of the Hamiltonian is

$$E_{\pm}(\mathbf{k}) = \pm\sqrt{\Delta(\mathbf{k})} = \pm\sqrt{h_x^2(\mathbf{k}) + h_y^2(\mathbf{k}) + h_z^2(\mathbf{k})}. \quad (\text{R8})$$

The emergence of the exceptional point \mathbf{k}_{EP} requires that

$$\Delta(\mathbf{k}_{\text{EP}}) = h_x^2(\mathbf{k}_{\text{EP}}) + h_y^2(\mathbf{k}_{\text{EP}}) + h_z^2(\mathbf{k}_{\text{EP}}) = 0 \quad (\text{R9})$$

and there exists an invertible matrix P such that

$$P^{-1}\mathcal{H}(\mathbf{k}_{\text{EP}})P = \begin{pmatrix} 0 & a \\ 0 & 0 \end{pmatrix} \quad (\text{R10})$$

with $a \neq 0$ a general complex number. The topological charge of the exceptional point is

$$\nu(\mathbf{k}_{\text{EP}}) = \frac{1}{2\pi i} \oint_{\Gamma(\mathbf{k}_{\text{EP}})} d\mathbf{k} \cdot \nabla_{\mathbf{k}} \ln \Delta(\mathbf{k}). \quad (\text{R11})$$

which describes the winding number of the Bloch spectrum around $E_0 = E_{\pm}(\mathbf{k}_{\text{EP}}) = 0$ when \mathbf{k} winds around the \mathbf{k}_{EP} . Thus one can imagine that when $\nu(\mathbf{k}_{\text{EP}}) \neq 0$, the corresponding spectral area must be nonzero. Here we note that the above derivation does not require that the degeneracy point is an exceptional point. Actually, any degeneracy point with nonzero topological charge means the nonzero spectral area.

Now we consider the two-dimensional system with stable exceptional points (that is $\nu(\mathbf{k}_{\text{EP}}) = \pm 1$). The dispersion around the exceptional point \mathbf{k}_{EP} can be expanded [6] as

$$E_{\pm}(\mathbf{q}) = \pm \sqrt{c_x q_x + c_y q_y} + O(|\mathbf{q}|), \quad (\text{R12})$$

where c_x, c_y are nonzero complex numbers and $c_{x/y} = c_{x/y}^r + i c_{x/y}^i$ with the superscript r/i indicates the real/imaginary part. The above equation implies that

$$\Delta(\mathbf{q}) \simeq c_x q_x + c_y q_y = c_x \tilde{\Delta}(\mathbf{q}) = c_x (q_x + c_1 q_y) \quad (\text{R13})$$

with $c_1 = c_y/c_x = c_1^r + i c_1^i$, where $\tilde{\Delta}(\mathbf{q}) = \text{Re} \tilde{\Delta}(\mathbf{q}) + i \text{Im} \tilde{\Delta}(\mathbf{q}) = (q_x + c_1^r q_y) + i c_1^i q_y$. Putting this equation into the topological charge formula Eq.(R11), equivalently, one can obtain that

$$\nu(\mathbf{k}_{\text{EP}}) = \text{sgn}(\det \begin{bmatrix} \partial_{q_x} \text{Re} \tilde{\Delta}(\mathbf{q}) & \partial_{q_y} \text{Re} \tilde{\Delta}(\mathbf{q}) \\ \partial_{q_x} \text{Im} \tilde{\Delta}(\mathbf{q}) & \partial_{q_y} \text{Im} \tilde{\Delta}(\mathbf{q}) \end{bmatrix}) = \text{sgn}(\det \begin{bmatrix} 1 & c_1^r \\ 0 & c_1^i \end{bmatrix}) = \text{sgn}(c_1^i), \quad (\text{R14})$$

where $\text{sgn}(c_1^i)$ represents the sign of the c_1^i . Note that in our statement in the ‘‘Corollary’’ section of the main text, $c_0 = \sqrt{c_x}$ and $c_1 = c_y/c_x$. Therefore, from Eq.(R14), a stable EP ($\nu(\mathbf{k}_{\text{EP}}) = \pm 1$) requires a nonzero imaginary part of c_1 , which further ensures a nonzero spectral area A_{\pm} .

 In the revised version, we have made the following modifications

- In the main text, we have modified the sentence ‘‘Here c_0, c_1 are nonzero complex numbers and $c_1 \notin \mathbb{R}$.’’ in the first paragraph of the ‘‘Corollary’’ section as ‘‘Here c_0, c_1 are nonzero complex numbers and the stable exceptional point ensures the nonzero imaginary part of c_1 .’’
- We have added the above discussion into Supplementary Note 5-C to support the statements in the main text.

‘‘(12) On the right part of page 4, it is written, ‘‘As mentioned in the previous discussion, the appearance of the skin effect can be reflected in the dynamical properties.’’ However, the relationship between dynamical properties of the system and the skin effect is not mentioned anywhere in the previous section. ’’

In the revised version, we have modified the statement as

‘‘So far, we have shown the features of the energy spectrum and wave function in the system with GDSE. We expect some observable phenomena from the skin effect, which motivates us to examine the dynamical properties for the photonic crystal model in Eq.(4). ’’

‘‘(13) ‘‘Lei algebra’’. ’’

We have corrected this typo in the revised version. Many thanks to our reviewer.

-
- [1] Kawabata, K., Shiozaki, K., Ueda, M. & Sato, M. Symmetry and Topology in Non-Hermitian Physics. *Phys. Rev. X* **9**, 041015 (2019).
 [2] Okuma, N., Kawabata, K., Shiozaki, K. & Sato, M. Topological Origin of Non-Hermitian Skin Effects. *Phys. Rev. Lett.* **124**, 086801 (2020).
 [3] Kawabata, K., Okuma, N. & Sato, M. Non-Bloch band theory of non-Hermitian Hamiltonians in the symplectic class. *Phys. Rev. B* **101**, 195147 (2020).
 [4] Hofmann, T. *et al.* Reciprocal skin effect and its realization in a topoelectrical circuit. *Phys. Rev. Research* **2**, 023265 (2020).
 [5] Yang, Z., Schnyder, A. P., Hu, J. & Chiu, C.-K. Fermion Doubling Theorems in Two-Dimensional Non-Hermitian Systems for Fermi Points and Exceptional Points. *Phys. Rev. Lett.* **126**, 086401 (2021).
 [6] Shen, H., Zhen, B. & Fu, L. Topological Band Theory for Non-Hermitian Hamiltonians. *Phys. Rev. Lett.* **120**, 146402 (2018).

REVIEWERS' COMMENTS

Reviewer #1 (Remarks to the Author):

I would like to thank the authors for their great efforts to answer my questions and comments. Although there are a couple of points which cannot be proved mathematically rigorously at the moment, most of my previous concerns are clearly resolved by the authors in their response letter and revised manuscript. Now I can recommend the revised manuscript for publication in Nature Communications in its current form.

Reviewer #2 (Remarks to the Author):

The revised manuscript, with more detailed results and discussions included, has been improved much. I think that all previously raised issues except for minor details have been satisfactorily addressed, and I can recommend the publication of the manuscript once the following small points have been addressed:

- 1) There are still grammatical errors in several sentences, for example in the third sentence of the third paragraph on page 1 it is stated "Many a novel phenomenon related to exceptional points has been predicted and observed [...]", which should probably be "Many novel phenomena related to exceptional points have been predicted and observed [...]".
- 2) I apologize for being pedantic about this, but the colorbars and -maps in Fig. 3 (d) for the two triangles should be combined into a single one, which would reveal that the localization almost entirely concentrates on the lower right corner. The same holds for Fig. 3 (h) with regards to the use of a single colorbar and -map.
- 3) Along these lines: The CSE is stated to localize almost all eigenstates at the corner, but the colorbar of Fig. 3 (c) reveals that we are far from $W(x) = 1$. Adding a similar scaling analysis as for the GDSE in Supplementary Note 4 B.4 would be helpful to avoid misinterpretations.
- 4) The eigenstates used for calculating $W(x)$ are probably right eigenstates and should be stated as such.

Reviewer #3 (Remarks to the Author):

The author has revised the manuscript precisely to address all my questions and concerns. In particular, the revised manuscript has no logical leaps, and the terminologies are properly defined. As I mentioned in my previous report, I think the results in the manuscript are both timely and impactful, and I would recommend publication as it stands.